EMBO
Molecular Medicine

# Fertility protection during chemotherapy treatment by boosting the NAD(P)$^+$ metabolome

Wing-Hong Jonathan Ho[1,2,3,12], Maria B Marinova[1,2,12], Dave R Listijono[1,2], Michael J Bertoldo[1,2], Dulama Richani[2], Lynn-Jee Kim[1], Amelia Brown[2], Angelique H Riepsamen[2], Safaa Cabot[1], Emily R Frost [ID][2], Sonia Bustamante [ID][4], Ling Zhong[4], Kaisa Selesniemi[5], Derek Wong[1], Romanthi Madawala[1], Maria Marchante [ID][6,7], Dale M Goss[1], Catherine Li[1], Toshiyuki Araki [ID][8], David J Livingston[9], Nigel Turner[1,10], David A Sinclair [ID][5], Kirsty A Walters[2], Hayden A Homer [ID][2,11], Robert B Gilchrist [ID][2,13] & Lindsay E Wu [ID][1,13][✉]

## Abstract

**Chemotherapy induced ovarian failure and infertility is an important concern in female cancer patients of reproductive age or younger, and non-invasive, pharmacological approaches to maintain ovarian function are urgently needed. Given the role of reduced nicotinamide adenine dinucleotide phosphate (NADPH) as an essential cofactor for drug detoxification, we sought to test whether boosting the NAD(P)$^+$ metabolome could protect ovarian function. We show that pharmacological or transgenic strategies to replenish the NAD$^+$ metabolome ameliorates chemotherapy induced female infertility in mice, as measured by oocyte yield, follicle health, and functional breeding trials. Importantly, treatment of a triple-negative breast cancer mouse model with the NAD$^+$ precursor nicotinamide mononucleotide (NMN) reduced tumour growth and did not impair the efficacy of chemotherapy drugs in vivo or in diverse cancer cell lines. Overall, these findings raise the possibility that NAD$^+$ precursors could be a non-invasive strategy for maintaining ovarian function in cancer patients, with potential benefits in cancer therapy.**

**Keywords** Oncofertility; Nicotinamide Adenine Dinucleotide (NAD$^+$); Nicotinamide Mononucleotide (NMN); Ovarian Toxicity; Infertility
**Subject Categories** Cancer; Metabolism; Urogenital System

## Introduction

The improved survival of cancer patients has led to an increasing need for strategies that protect ovarian function and the finite ovarian reserve from commonly used cytotoxic chemotherapy (Bertoldo et al, 2020b; Woodruff, 2009). While effective in oncology, these agents can lead to long-term adverse health outcomes for female cancer patients, including endocrine disruption, amenorrhea, premature ovarian insufficiency and infertility (Green et al, 2009a; Hudson, 2010; Morgan et al, 2012), with life-long health, psychological and social consequences (Gorman et al, 2015; Hudson et al, 2013). Aside from the devastating social and health consequences of infertility and its impacts on family planning (Bertoldo et al, 2020b), this can inflict damage to health through accelerated ovarian ageing and premature ovarian insufficiency, with early menopause symptoms due to estrogen insufficiency. In women of reproductive age or younger, options for preserving future fertility focus on the cryopreservation of reproductive material including oocytes, embryos, and in the case of child and adolescent cancer patients, ovarian tissue biopsies (Bertoldo et al, 2020b). These strategies must be undertaken prior to initiating chemotherapy, which may not be an option depending on the urgency of cancer treatment.

The best chance of maintaining future fertility options prior to chemotherapy treatment is the cryopreservation of oocytes or embryos following superovulation and IVF, however, this protocol can be too lengthy to delay chemotherapy and has a limited success rate. Moreover, superovulation followed by oocyte freezing is not suitable for childhood cancer patients (Deli et al, 2019). While these cryopreservation strategies provide options for the future ability to conceive genetic offspring, they do not prevent premature ovarian and endocrine failure which can lead to other long-term adverse

[1]School of Biomedical Sciences, UNSW Sydney, Kensington, NSW 2052, Australia. [2]School of Clinical Medicine, UNSW Sydney, Kensington, NSW 2052, Australia. [3]The Kinghorn Cancer Centre, St. Vincent's Hospital, Darlinghurst, NSW, Australia. [4]Bioanalytical Mass Spectrometry Facility, Mark Wainwright Analytical Centre, UNSW Sydney, Kensington, NSW 2052, Australia. [5]Paul F Glenn Laboratories for the Biological Mechanisms of Aging, Harvard Medical School, Boston, MA, USA. [6]IVI Foundation, Valencia, Spain. [7]Department of Pediatrics, Obstetrics and Gynaecology, Faculty of Medicine, University of Valencia, Valencia, Spain. [8]Department of Peripheral Nervous System Research, National Institute of Neuroscience, National Center of Neurology and Psychiatry, 4-1-1 Ogawa-higashi, Kodaira, Tokyo 187-8502, Japan. [9]Metro International Biotech, Worcester, MA, USA. [10]Victor Chang Cardiac Research Institute, Darlinghurst, NSW 2010, Australia. [11]Christopher Chen Oocyte Biology Laboratory, University of Queensland Centre for Clinical Research, Royal Brisbane & Women's Hospital, Herston, QLD 4029, Australia. [12]These authors contributed equally: Wing-Hong Jonathan Ho, Maria B Marinova. [13]These authors jointly supervised this work: Robert B Gilchrist, Lindsay E Wu. [✉]E-mail: lindsay.wu@unsw.edu.au

outcomes including endocrine failure, osteoporosis, metabolic disease and neurocognitive disorders (Bruning et al, 1990; Jeanes et al, 2007). In addition, many cancer patients do not have timely access to the advanced medical care needed for such fertility preservation procedures (Letourneau et al, 2012b). Concurrent treatment of the GnRH agonist goserelin with chemotherapy provides some benefits to fertility preservation in breast cancer patients (Moore et al, 2015), however, these effects are moderate and the interpretation of these data is complicated by differences in the proportion of women seeking pregnancy (Oktay et al, 2015). What is needed is a non-invasive pharmacological strategy to protect ovarian function and fertility from the damaging effects of chemotherapy.

In addition to infertility, cancer survivors face an increased risk of chronic disorders including metabolic disease, cardiovascular disease, neurological disorders, bone disorders and cancers unrelated to the initial diagnosis (Hudson et al, 2013). Together, this diverse constellation of disorders resembles a form of biological ageing, presenting a challenge to maintaining the long-term health of this patient cohort. Several interventions previously shown to slow biological ageing have been shown to prevent the acceleration in ovarian decline during chemotherapy treatment. For example, the mTOR inhibitor rapamycin, which is a robust pharmacological intervention for extending lifespan in model organisms (Lamming et al, 2012), also prevents chemotherapy-induced infertility (Goldman et al, 2017). Similarly, inhibition of PI3K/Akt signalling, which extends lifespan in model organisms, also prevents chemotherapy-induced ovarian depletion (Kalich-Philosoph et al, 2013). These results indicate that other longevity interventions might also delay chemotherapy induced infertility, particularly if they have previously been shown to slow or reverse ovarian ageing. Recently, we (Aflatounian et al, 2022; Bertoldo et al, 2020a; Campbell et al, 2022; Habibalahi et al, 2022) and others (Huang et al, 2022; Miao et al, 2020; Yang et al, 2020) showed that declining female fertility during biological aging was in part due to a decline in levels of the prominent redox cofactor nicotinamide adenine dinucleotide (NAD$^+$). Reversal of this age-associated decline using the NAD$^+$ precursor nicotinamide mononucleotide (NMN) restored functional fertility, even when treatment was started at a post-reproductive age. More recently, others have found elevated expression of the NAD$^+$ glycohydrolase enzyme CD38 in the ovary during reproductive ageing, and showed that blocking the degradation of NAD$^+$ through small molecule inhibition of CD38 can prolong female reproductive lifespan (Perrone et al, 2023; Yang et al, 2024).

NAD$^+$ plays another key role in slowing ovarian function by serving as a co-substrate of poly-ADP-ribose polymerase (PARP) enzymes in the DNA damage response (Durkacz et al, 1980), which is important to chemotherapy induced infertility (Nguyen et al, 2019). These enzymes can be among the greatest consumers of NAD$^+$ in the cell, depleting cellular NAD$^+$ reserves to the point of impacting cell survival (Fang et al, 2016). Chemotherapy can also induce infiltration into the ovaries of immune cells (Du et al, 2022), which can express the NAD$^+$ consuming enzyme CD38, another determinant of ovarian ageing (Perrone et al, 2023; Yang et al, 2024). Further, many chemotherapy agents undergo drug metabolism by reductase enzymes that are fuelled by an NADPH cofactor, with NADP$^+$ and NADPH levels dependent on the availability of NAD$^+$ as a substrate for NAD$^+$ kinase. Considering the role of

NADPH-fuelled drug metabolism, NAD$^+$-consuming PARP enzymes in DNA repair following chemotherapy, and recent findings around the role of NAD$^+$ in reproductive ageing, we sought to test whether supporting NAD$^+$ homeostasis could impact female infertility caused by chemotherapy treatment. To address this, we used a series of orthogonal approaches to test the role of NAD(P)$^+$ homeostasis in chemotherapy induced infertility, including treatment with the NAD$^+$ precursor NMN, along with transgenic overexpression of the NAD$^+$ biosynthetic enzymes nicotinamide mononucleotide adenyltransferase 1 and 3 (NMNAT1 and NMNAT3). We find that these interventions can partially prevent a loss of ovarian function and functional fertility caused by the anthracycline chemotherapy drug doxorubicin (Dox), and importantly, that this protection against infertility does not come at the cost of reduced efficacy of chemotherapy against cancer cell lines in vitro or tumour models in vivo.

## Results

To test whether NAD(P)$^+$ homeostasis is involved in chemotherapy-induced ovarian toxicity, we first used treatment with nicotinamide mononucleotide (NMN), an orally bioavailable metabolic precursor to NAD$^+$ and NADP$^+$ (Yoshino et al, 2011). In our experimental model, 8-week-old female C57BL6 mice were treated with or without chemotherapy, with or without the co-administration and ongoing treatment of NMN, for a 2 × 2 study design (Fig. 1A). Animals treated with NMN received a single injection at the time of chemotherapy, in addition to ongoing treatment with NMN in drinking water at 2 g/L, which started 72 h prior to chemotherapy and continued until the end of all experiments. These included measures of oocyte yield, ovarian reserve, follicle health and fertility. Functional readouts of ovarian function and fertility were from 8 weeks (56 days) following chemotherapy, by comparison, the minimum period needed for primordial follicles to develop to the antral stage or ovulation in mice has been estimated from between 20 days (Pedersen, 1970), 42 days (Oakberg, 1979) to 47 days (Zheng et al, 2014), with a recent estimate of 14–16 days for the primary follicle transition (Richard et al, 2023).

We first tested this in animals treated with the anthracycline chemotherapy drug, doxorubicin (Dox), which is associated with an increased risk of ovarian failure in patients (Green et al, 2009a; Green et al, 2009b; Letourneau et al, 2012a; Lipshultz et al, 2017) and which causes direct oocyte toxicity, ovarian toxicity and infertility in mice (Ben-Aharon et al, 2010; Lopes et al, 2020; Perez et al, 1997; Wang et al, 2019). Dox was used at 10 mg/kg, which was chosen based on previous work demonstrating lasting changes to the ovarian reserve (Ben-Aharon et al, 2010). Animals were treated with Dox and/or NMN, and 8 weeks later were stimulated with pregnant mare's serum gonadotrophin (PMSG), with the number of germinal vesicle (GV) stage cumulus oocyte complexes (COCs) recovered from ovarian antral follicles used as a readout for oocyte yield (Fig. 1B). As expected, Dox caused a decrease in COC yield, however, this was rescued by NMN co-treatment (Fig. 1B). To test whether this was unique to this single chemotherapy agent, we also repeated the same experimental design in animals that were instead treated with cisplatin (CDDP, 5 mg/kg), which is also associated with ovarian toxicity (Bildik et al, 2015; Lopes et al, 2020;

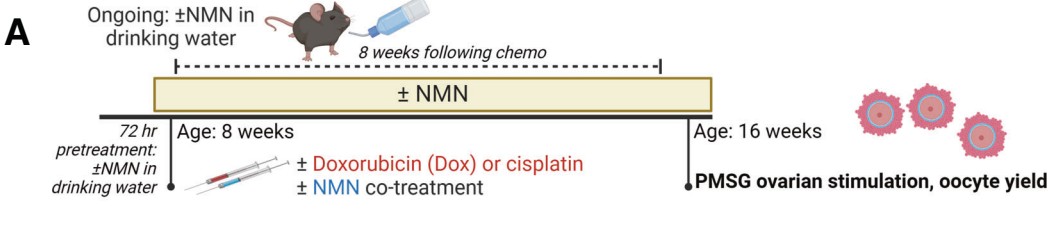

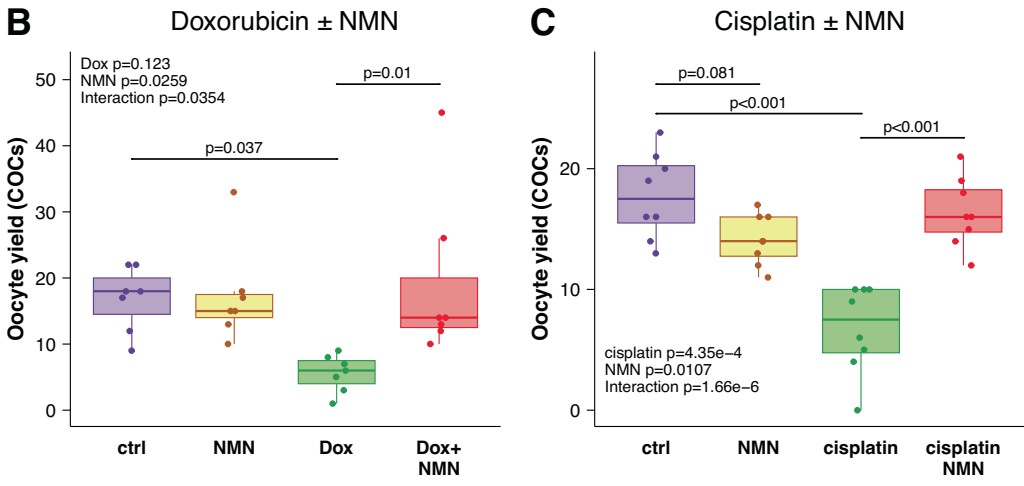

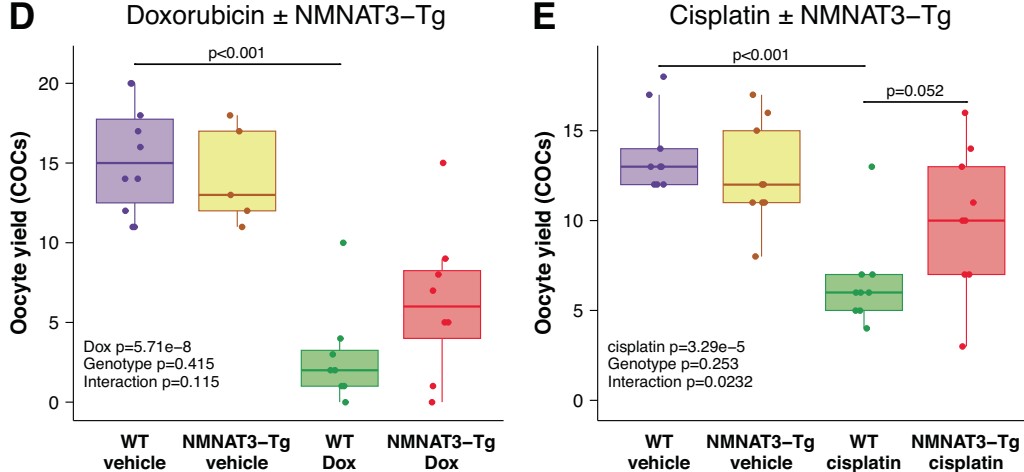

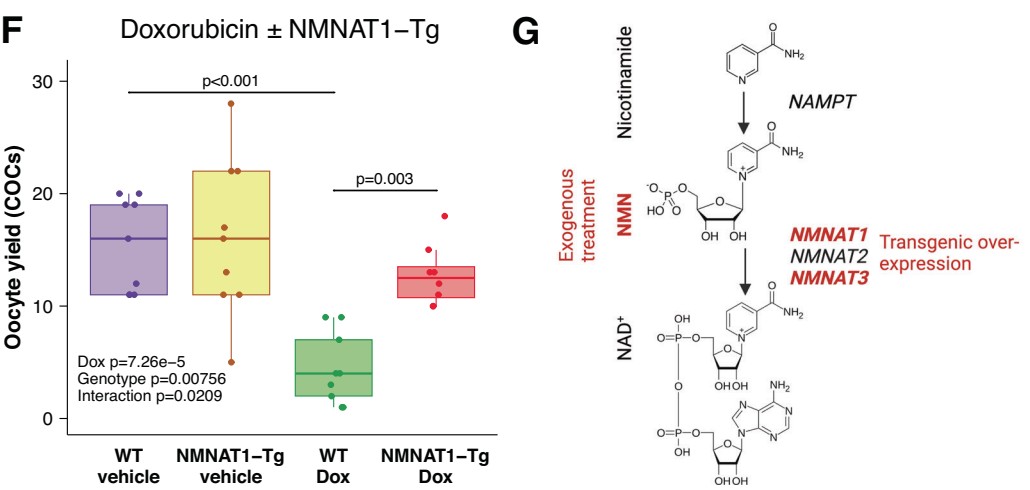

◀ **Figure 1.  NMN treatment and transgenic overexpression of NAD⁺ biosynthetic enzymes attenuates the loss of oocyte yield following chemotherapy treatment.**

(A) Female mice were treated with or without the NAD⁺ precursor nicotinamide mononucleotide (NMN) in drinking water (2 g/L) starting from 72 h prior to an i.p. injection of doxorubicin (Dox, 10 mg/kg) with or without an additional i.p. bolus of NMN (200 mg/kg). Eight weeks later, animals were stimulated with PMSG to determine the production of cumulus oocyte complexes (COCs), with separate cohorts used for histology to determine follicle numbers and health, and for breeding trials to determine fertility. (B) Wild-type females treated with or without NMN and the chemotherapy agent doxorubicin (Dox). An identical experiment was performed in transgenic animals over-expressing the NAD⁺ biosynthetic enzymes (C) NMNAT3 and (D) NMNAT1. This experiment was also repeated with cisplatin treatment in combination with (E) NMN treatment and (F) NMNAT3 transgenic mice. (G) Summary of strategies used to elevate NAD⁺ production, including exogenous treatment with NMN, and transgenic overexpression of the NAD⁺ biosynthetic enzymes NMNAT1 and NMNAT3. P-values from Bonferroni-adjusted t-tests derived from estimated marginal means analysis of a linear model of Dox and NMN treatment. n = 5–9 per group as indicated by the number of dots indicating data from separate animals. Data in this and all subsequent figures are summarised as Tukey boxplots, showing the 25–75% interquartile range with whiskers indicating 95% confidence intervals, mean values indicated by a line within boxplots.

Nguyen et al, 2019). This dose was chosen based on previous work, which established a reduction in follicle counts and demonstrated DNA damage in adult mice (Nguyen et al, 2019). In addition to its ovarian toxicity, cisplatin was chosen due to its widespread use in cancer treatment, and the previously described impacts of platinum drugs on the NAD metabolome (Guan et al, 2017; Kim et al, 2014). As with the previous result, NMN co-treatment also protected against a loss in oocyte yield caused by cisplatin (Fig. 1C). To further validate these findings using an orthogonal approach, rather than NMN treatment, we next used transgenic mice that globally over-expressed the NAD⁺ biosynthetic enzymes NMNAT3 (Fig. 1D,E), which localises to mitochondria, and NMNAT1 (Fig. 1F), which localises to the nucleus (Yahata et al, 2009). These enzymes catalyse the final step of NAD⁺ biosynthesis, converting NMN into NAD⁺ (Fig. 1G). $Nmnat3^{Tg/+}$ animals were partially protected against a loss of oocytes caused by cisplatin, but not DOX (Fig. 1D,E), while oocyte numbers in $Nmnat1^{Tg/+}$ animals were protected against Dox (Fig. 1F). To assess the meiotic competence of these oocytes, we measured their meiotic progression through germinal vesicle breakdown (GVBD) and polar body extrusion (PBE) (Fig. EV1). While meiotic progression was always lowest in animals that received chemotherapy alone, this was not statistically significant, and we cannot conclude that chemotherapy impacts oocyte maturation. Regardless, chemotherapy, and its rescue by NMN, impacts follicle survival, as evidenced by COC yield (Fig. 1). Together, these experiments support the concept that NAD⁺ levels could be a target to maintain oocyte numbers in the context of chemotherapy treatment.

To test whether this increased COC yield translates to changes in fertility, two separate breeding trials were undertaken, where animals were treated with or without Dox and/or NMN (Fig. 2; (n = 10/group), or with or without cisplatin and/or NMN (Fig. 3; (n = 10/group), with the number of live offspring born used as readouts for functional fertility (Figs. 2 and 3). The Dox breeding trial (Fig. 2A) started at 8 weeks (56 days) and continued for over 250 days after chemotherapy, which is in excess of the ~47–50 days required for a full cycle of follicle development in sexually mature adult mice (Zheng et al, 2014), the uppermost limit from estimates that range down to 20 days (Oakberg, 1979; Pedersen, 1970). As expected, Dox caused a notable reduction in the overall number of pups born during the breeding trial (Fig. 2B–G), with NMN co-treatment with Dox partially protecting against this decrease in reproductive output. Figure 2B presents these data as the cumulative number of pups per group as a function of time, which is relevant to interpreting potential impacts on the primordial reserve versus the impacts on growing follicles, with similar data

showing reproductive output from individual animals in each group shown in Fig. EV2A. Notably, using a cut-off for mating that took place more than 150 days following treatment (i.e. ~3 waves of complete folliculogenesis), four of ten Dox+NMN treated animals had pregnancies that yielded 26 pups, compared to two of ten from the Dox alone group, which yielded only 4 pups (Figs. 2C and EV2A). Further, these data are also shown as a function of cumulative breeding output over successive mating cycles (Fig. 2C), which were confirmed by the presence of a vaginal plug following timed mating which allowed correlation of successful copulation with live birth outcomes—individual data shown in Fig. EV2B. In line with the overall difference in the number of pups born throughout the entire breeding trial (Fig. 2B–D), the number of pups born per round of copulation per female (Fig. 2F) and the number of live pups per litter (Fig. 2G) were also partially rescued in animals that received both Dox and NMN, compared to those receiving Dox alone. Interestingly, the number of pups in each litter was increased by NMN treatment even in the absence of Dox treatment (Fig. 2G), when compared to the untreated controls, which is in line with our previous work (Bertoldo et al, 2020a). While this study used n = 10 per group, some panels in Fig. 2 and Fig. EV2 show less than 10 datapoints in each group, due to a failure to achieve mating or pregnancy—notably in the Dox alone group.

Given these results, we next extended these breeding trial studies to a previously described model of cisplatin induced infertility (Gonfloni et al, 2009; Kerr et al, 2012), where the cisplatin (CDDP) insult is delivered to prepubertal mice which have a high primordial follicle reserve. In this model, 1-week-old pups were treated with or without CDDP (2 mg/kg) alone, and 2 weeks later, weaned onto drinking water supplemented with or without NMN (2 g/L) (Fig. 3A). As expected, CDDP adversely affected fertility (Figs. 3B–G and EV2C,D). Overall, NMN similarly ameliorated fertility loss in animals treated with CDDP (Fig. 3) as observed in the doxorubicin trial (Fig. 2). Animals receiving CDDP followed by NMN had more total pups/dam (Fig. 3B,E), more pups/mating event (Fig. 3C,F), with larger litters (Fig. 3D,G), than animals receiving CDDP without NMN. As with the Dox breeding trial, animals receiving CDDP and NMN had improved fertility potential for an extended period. At >100 days after CDDP treatment, 100% (9/9) of animals that received CDDP followed by NMN had litters, yielding 107 pups. By comparison, only 22% (2/9) of animals receiving CDDP alone had litters beyond this timepoint, yielding only 6 pups (Figs. 3B and EV2C). At 150 days post-chemo, 5/9 from the CDP+NMN group had litters, yielding 32 pups, compared to zero litters from the CDDP alone group (Fig. EV2C).

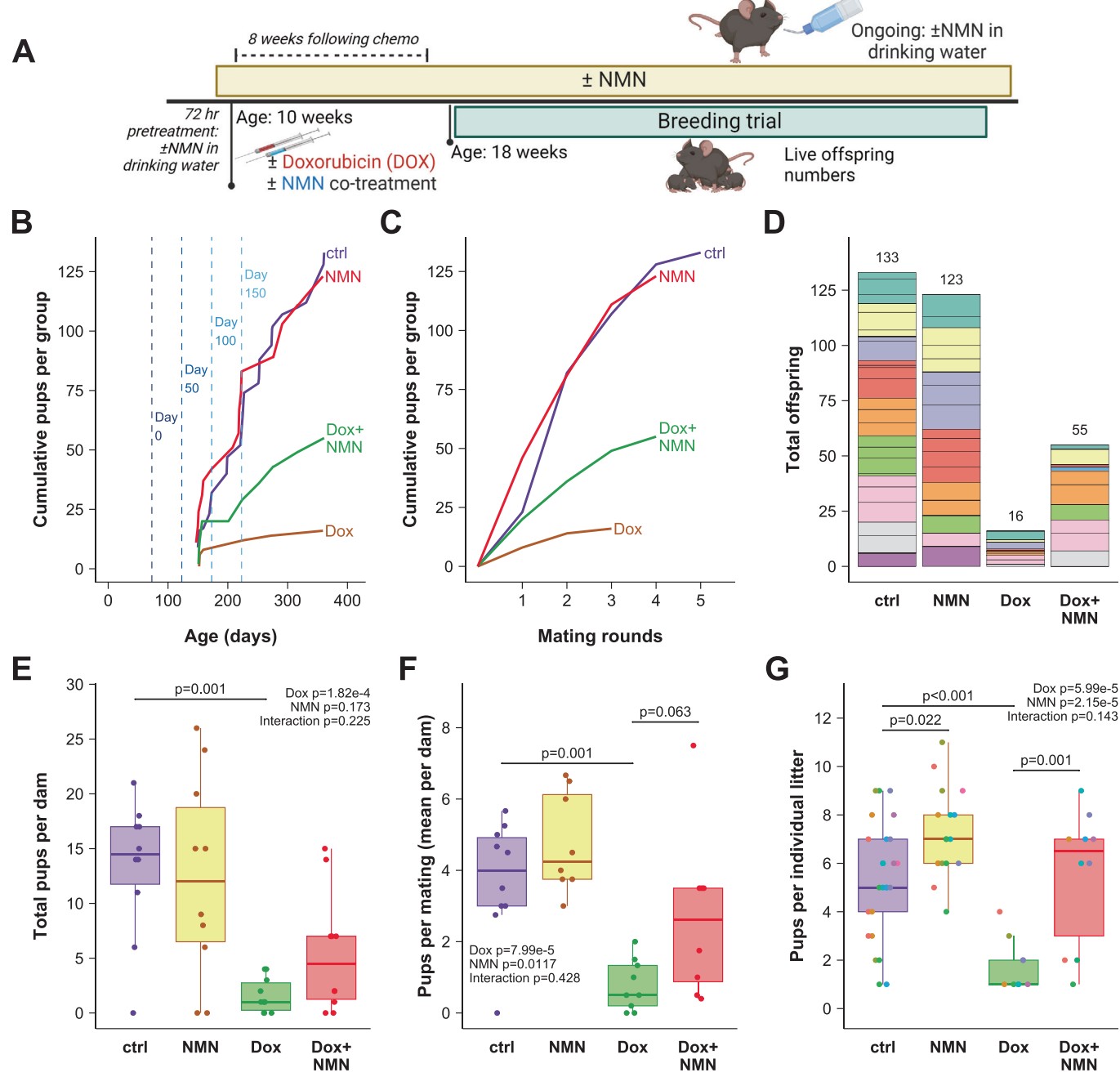

**Figure 2. NMN treatment ameliorates doxorubicin-induced infertility.**

(A) Animals were treated with doxorubicin (Dox) or vehicle control and/or NMN, and 8 weeks later were subject to mating trials to determine fertility. (B) Cumulative number of pups for each group over time, with the timing of Dox treatment indicated by a dark blue line, with lighter dashed lines for subsequent 50-day increments. Results are also expressed as (C) cumulative number of pups in each cohort per round of confirmed mating, as determined by the presence of vaginal plugs, (D) summarised as totals per group with stacked colours for each dam. (E) Total number of pups per dam. (F) Number of live offspring born per round of successful mating, which was determined by the presence of a vaginal plug following timed overnight mating. (G) Number of pups in each litter, datapoints coloured for each animal within that group. N = 10 mice per group, however, not all groups show 10 datapoints on all graphs where some animals failed to deliver pups and/or mate. P-values from Bonferroni-adjusted t-tests derived from estimated marginal means analysis of a linear model of Dox and NMN treatment. Tukey boxplots show the 25–75% interquartile range with whiskers indicating 95% confidence intervals, mean values indicated by a line within boxplots.

Notably, follicle maturation in mice occurs in two distinct waves, with the first wave, which lasts until ~3 months of age, maturing from primordial to antral stage in only 23 days, with the subsequent wave maturing in 47 days (Zheng et al, 2014). Unlike the previous breeding trial (Fig. 2) where Dox treatment was in adults, cisplatin was delivered at the prepubertal age of 7 days, during mobilisation of the first wave of follicles. These differences in breeding outcomes at 100 and 150 days represent around 3 and 4 cycles of follicle

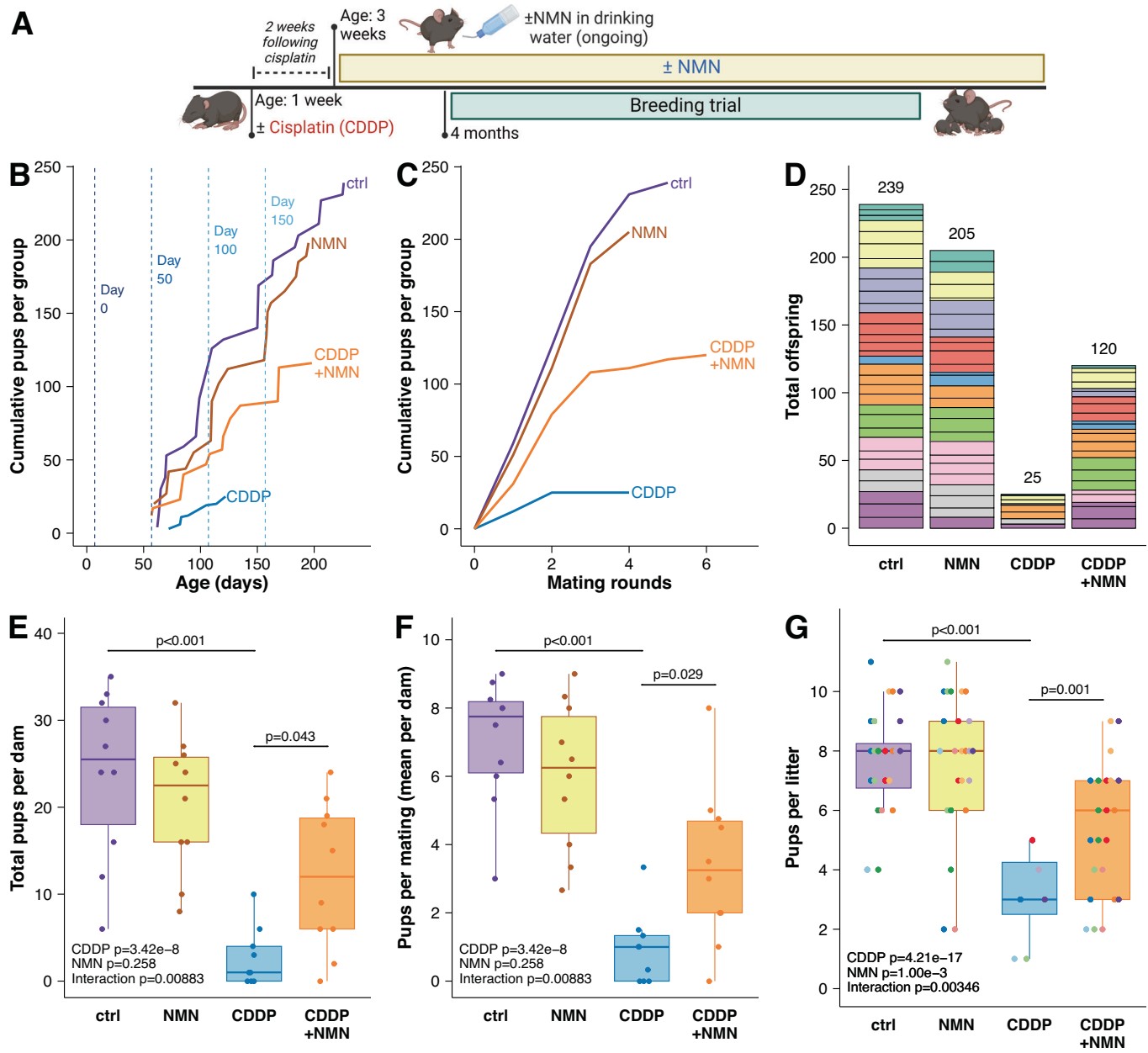

**Figure 3. NMN treatment following cisplatin (CDDP) ameliorates infertility.**

(A) Animals were treated with cisplatin (2 mg/kg) or saline control, without any NMN co-treatment. Two weeks later, animals received drinking water containing NMN (2 g/L) or untreated water, followed by a breeding trial from 4 months of age. (B) Cumulative number of pups for each group over time, with the timing of CDDP treatment indicated by a dark blue line, with lighter dashed lines for subsequent 50-day increments. Results are also expressed as (C) cumulative number of pups in each cohort per round of confirmed mating, as determined by the presence of vaginal plugs, (D) summarised as totals per group with stacked colours for each dam. (E) Total pups per dam. (F) Number of live offspring born per round of successful mating. (G) Number of pups in each litter, datapoints coloured for each animal within that group. N = 10 per group for control and NMN groups, n = 9 for CDDP and CDDP + NMN after one animal was identified as an outlier (ROUT cut-off Q = 0.1) in the CDDP group, and one animal from the CDDP + NMN group was removed prior to starting the breeding trial. P-values from Bonferroni-adjusted t-tests derived from estimated marginal means analysis of a linear model of Dox and NMN treatment. Tukey boxplots show the 25–75% interquartile range with whiskers indicating 95% confidence intervals, mean values indicated by a line within boxplots.

maturation, respectively. Further, cisplatin treatment was in prepubertal mice, when follicles are not yet growing. Together, this suggests that the benefits of NMN treatment are persistent, and are not restricted to the pool of follicles that were actively growing at the time of chemotherapy. Together, these data demonstrate

protection of ovarian function by NMN in the face of chemotherapy, as evidenced by oocyte yield (Fig. 1) and improved fertility (Figs. 2 and 3).

The changes in COC yield (Fig. 1) and fertility (Figs. 2, 3 and EV2) are likely to be mediated at the level of the ovary, via

treatment effects on ovarian reserve and/or follicle development. Chemotherapy has been described to deplete the ovarian reserve through both direct toxicity to oocytes (Nguyen et al, 2019), granulosa cell apoptosis (Lopes et al, 2020; Morgan et al, 2013; Yuksel et al, 2015) and follicle atresia leading to premature activation of the finite primordial follicle pool, leading to premature depletion of the ovarian reserve (Goldman et al, 2017; Kalich-Philosoph et al, 2013). To investigate effects on the ovarian reserve and follicle development, we treated animals with or without Dox +/− NMN (Fig. 4). Eight weeks later, animals were subjected to vaginal smearing for at least 10 days to track their cyclicity (Appendix Fig. S1). At diestrus, animals were euthanised and ovaries collected for detailed assessment of ovarian reserve and follicle health (Fig. 4). These ovaries were subjected to resin embedding, thick (20 μm) sectioning and stereology, with counting by a blinded investigator of all stages of follicle development in every third section across each entire ovary, including follicles that were assessed as healthy or unhealthy (Fig. 4). Morphological assessment included the classification of several unhealthy follicle types, described further in Methods, with examples shown in Fig. EV3, which are summed into a single "unhealthy" category. Briefly, unhealthy follicles included those which were atretic, with missing oocytes, zona pellucida remnants, enlarged oocytes with undifferentiated granulosa cells, multinuclear, vacuoles present, enlarged oocyte with undifferentiated GCs, biovular follicles, >10% pycnotic granulosa cells, damaged oocytes and others (Fig. EV3). As expected, Dox treatment caused a reduction in the primordial follicle reserve, which was not changed by NMN treatment (Fig. 4A). Of the remaining follicles that were assessed, there was, however, a stark difference in their health, with a four-fold increase in the number of unhealthy primordial follicles with Dox treatment (Fig. 4A). Notably, this increase in unhealthy primordial follicle numbers was restored by NMN co-treatment (Fig. 4A), with a similar, non-significant trend also observed in transitory (Fig. 4B) and primary (Fig. 4C) follicles, with a trend ($p = 0.091$) for an overall reduction in the number of unhealthy follicles of all stages across the entire ovary (Fig. 4D).

In addition to these differences in the health of the ovarian reserve, NMN co-treatment with Dox also led to an increase in the number of large antral follicles (Fig. 4E,F), with a similar trend in the number of corpora lutea (Fig. 4G). These findings would be in line with the increased oocyte yield observed in Fig. 1, as well as the increase in litter size observed in Figs. 2 and 3, and could suggest that NMN can improve the number of follicles that reach the antral stage and subsequent ovulation. The corpus luteum normally secretes progesterone, and while there was a trend towards reduced serum progesterone levels in animals treated with Dox only, this was not significant (Fig. 4H). Representative images of entire ovarian sections from each group are shown (Fig. 4I–L), and broadly reflect the blinded counts of antral follicle numbers. These representative images (Fig. 4I–L) suggest a reduction in overall size of the ovary following Dox treatment, which was also observed macroscopically (Fig. EV4A) and although the overall weights of ovaries showed a similar trend, this was not significant (Fig. EV4B). Given that NMN improved follicle health without impacting the ovarian reserve (Fig. 4), it is possible that the improved COC yield (Fig. 1) and fertility (Figs. 2 and 3) are due to its protection of granulosa cells and other somatic cell types in the ovary, which

would be in line with previous findings on the toxicity of Dox towards these cells (Morgan et al, 2013).

For exogenous NMN to protect the ovary, one assumption is that it must be bioavailable and able to elevate NMN in this tissue. To assess this, we repeated dosing with NMN and Dox and collected ovaries at an acute timepoint of 6 h following treatment, which were subjected to targeted mass spectrometry analysis of the $NAD^+$ metabolome (Fig. 5), including the metabolites nicotinamide (Fig. 5A), NMN (Fig. 5B), $NAD^+$ (Fig. 5C), $NADP^+$ (Fig. 5D), 1-methyl-nicotinamide (Fig. 5E), nicotinic acid adenine dinucleotide (Fig. 5F), NADH (Fig. 5G), and NADPH (Fig. 5H). The role of each of these metabolites in $NAD^+$ biosynthesis is summarised in Fig. 5I. These data confirmed that exogenous NMN treatment elevated ovarian NMN, replenishing its levels following its depletion by Dox treatment (Fig. 5B). As expected, exogenous NMN treatment increased levels of the excretion product 1-methyl-nicotinamide (Fig. 5E), likely reflecting the excretion of excess $NAD^+$ precursors. One rationale for this investigation was that cytotoxic chemotherapy drugs would deplete the $NAD^+$ metabolome due to their activation of $NAD^+$ consuming PARP enzymes. While there was a trend towards declining $NAD^+$ levels with Dox treatment (Fig. 5C), this effect did not reach significance ($p = 0.069$), and nor was there an impact of either NMN or Dox treatment on the reduced equivalent NADH (Fig. 5G). Given the critical role of this cofactor in cell survival, it is likely that $NAD^+$ levels are tightly defended by upregulating $NAD^+$ biosynthesis, depleting levels of precursors such as NMN (Fig. 5B). There was a strong interaction ($p = 0.01$) between Dox and NMN treatment on NADPH (Fig. 5H), where a reduction in NADPH with Dox treatment was rescued by NMN co-treatment. This is consistent with well-characterised pathways for Dox metabolism, all of which are powered by an NADPH cofactor (Fig. 5J). This includes the one-electron reduction of Dox into its semiquinone form by P450 reductase enzymes, two-electron reduction into the alcohol metabolite doxorubicinol by carbonyl reductase enzymes, reduction into doxorubicin 7-deoxyaglycone by the action of cytosolic NADPH-dependent glycosidases, and hydrolysis by NADPH-dependent microsomal oxidoreductases into doxorubicin hydroxyaglycone (Fig. 5J). Every single one of these drug metabolites is dependent upon NADPH consumption, explaining its decline with Dox treatment. In addition to this, the reactive doxorubicin semiquinone radical can be recycled back to doxorubicin, generating an $\cdot O_2^-$ superoxide that leads to reactive $H_2O_2$ formation, which is neutralised into $H_2O$ by the conversion of reduced glutathione (GSH) into its oxidised form (GSSG). The glutathione system is also likely to be active in this scenario to neutralise other sources of cellular ROS that are generated by Dox treatment. The regeneration of oxidised GSSG into reduced GSH by glutathione reductase is again powered by NADPH, further depleting its levels. Together, these known pathways for Dox detoxification would explain the reduction in NADPH levels (Fig. 5H). Given that $NADP^+$ is generated by an NAD kinase that uses $NAD^+$ as a substrate, providing exogenous NMN could allow for increased flux from tightly defended $NAD^+$ levels into NADP(H) production (Fig. 5I). In line with this, Dox treatment and NMN treatment had opposing effects on ovarian NMN (Fig. 5B), with a reduction in NMN during Dox treatment and an increase in ovarian NMN with exogenous NMN treatment. This reduction in NMN with Dox (Fig. 5B) was most likely due to its

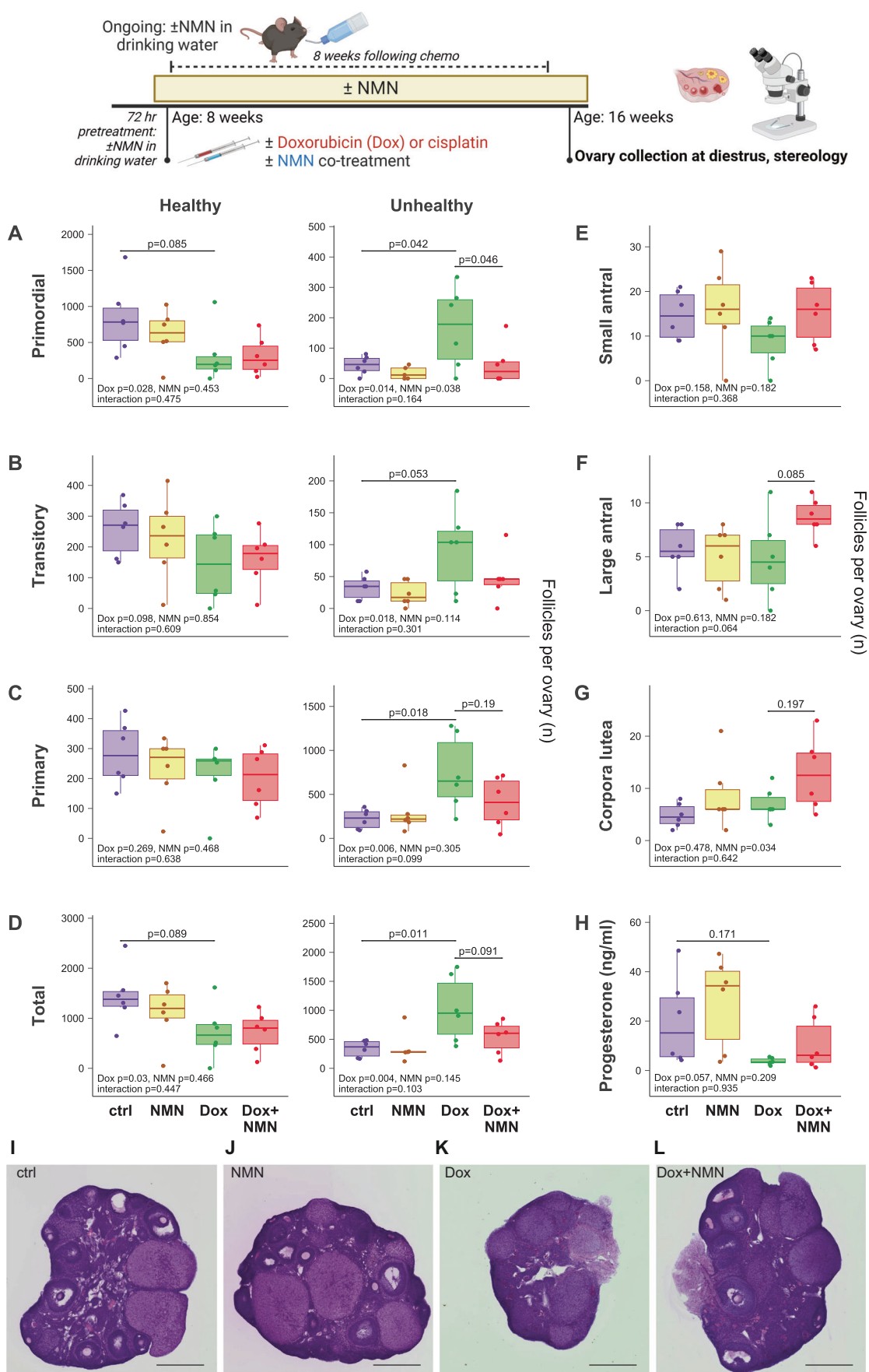

**Figure 4. Doxorubicin impairs follicle health and is rescued by NMN at the primordial stage.**

Females were treated with doxorubicin (Dox) and/or NMN as per Fig. 1A. Eight weeks later, animals were euthanased at the diestrus stage and ovaries collected for resin embedding, staining and stereology to assess follicle reserve and morphological health in (A) primordial, (B) transitory, (C) primary and (D) total follicles. Total numbers of (E) small and (F) large antral follicles were counted, along with (G) corpora lutea, which are involved in (H) progesterone secretion. (I–L) Representative sections of ovaries from (I) control, (J) NMN, (K) Dox and (L) Dox and NMN co-treated animals, scale bar is 500 µm. $n = 6$ per group. Follicles were counted from every third section across the entire ovary. *P*-values from Bonferroni-adjusted t-tests derived from estimated marginal means analysis of a linear model of Dox and NMN treatment. Tukey boxplots show the 25–75% interquartile range with whiskers indicating 95% confidence intervals, mean values indicated by a line within boxplots.

flux into NADP(H) production, where NADPH is required by xenobiotic detoxification enzymes for the metabolism of Dox.

Although NADPH is required for the metabolism and eventual detoxification and excretion of Dox, the immediate products of NADPH-fuelled Dox metabolism include doxorubicinol (Fig. 5J), which is responsible for the cardiotoxicity of Dox (Olson et al, 1988). One conceptual risk of restoring NADPH (Fig. 5H) through providing NMN could be an elevated conversion of Dox into doxorubicinol and enhanced cardiotoxicity. To address this, we tested NMN in a model of Dox-induced cardiotoxicity in mice, where animals were co-treated with or without NMN in the same $2 \times 2$ design, and cardiac function was assessed by µ-ultrasound imaging (Fig. EV5). We observed no exacerbation of Dox-induced cardiotoxicity, with a trend towards reduced changes in cardiac function during NMN treatment (Fig. EV5), which is consistent with recent findings using other strategies to elevate NAD$^+$ or prevent its breakdown by CD38 (Margier et al, 2022; Peclat et al, 2024; Zheng et al, 2019). Together, these data and other findings suggest that if NMN treatment leads to increased carbonyl reductase activity, this does not translate into an increased risk of cardiac dysfunction.

To obtain a broader view of molecular changes in ovarian function that could mediate these changes, we conducted label-free quantitative proteomics analysis (Fig. 6) of whole ovaries that were collected at six hours following doxorubicin, using the contralateral ovaries from animals used for metabolomics in Fig. 5. As expected, Dox treatment led to substantial changes in the proteome (Fig. 6A,B). Within Dox treatment, a subset of proteins were altered by NMN co-administration (Fig. 6C), normalising protein levels back to those of non-Dox treated animals—highlighted by cluster 4 of the heatmap (Fig. 6A), and select proteins shown in Fig. 6E–P. Importantly, acute NMN treatment on its own did not significantly alter any proteins when compared to untreated controls (Fig. 6D). This could suggest that any benefit provided by NMN treatment would occur in the context of rescuing a challenged state, such as from chemotherapy treatment or biological ageing (Bertoldo et al, 2020a), rather than altering the underlying baseline physiology of the ovary. Doxorubicin treatment led to striking reductions in proteins involved in ceramide biosynthesis, including ceramide synthase 5 (Cers5) and ceramide transporter 1 (Cert1), which were rescued by NMN co-treatment (Fig. 6G,H). Ceramide biosynthesis feeds into the production of sphingosine-1-phosphate (S1P), with the balance between ceramide and S1P levels acting as a determinant of apoptosis in oocytes (Morita et al, 2000). Notably, treatment with S1P can prevent the loss of primordial follicles caused by chemotherapy (Li et al, 2014) and irradiation (Paris et al, 2002). Ceramide biosynthesis involves the conversion of serine and palmitoyl-CoA to 3-dehydrosphinganine, which is a substrate for the enzyme 3-dehydrosphinganine reductase. This enzyme requires NADPH as a cofactor to produce

sphinganine, which is the direct substrate of ceramide synthase (CERS) enzymes including ceramide synthase 5. Given that NADPH is reduced by Dox treatment (Fig. 5H) due to its likely consumption in Dox metabolism (Fig. 5J), one possibility is that decreased activity of the NADPH-dependent enzyme 3-dehydrosphinganine reductase leads to decreased levels of the CERS5 substrate sphinganine, with decreased substrate availability (Fig. 5C) resulting in downregulation of this enzyme (Fig. 6G). Similarly, decreased ceramide production could also lead to downregulation of its transporter, CERT1 (Fig. 6H).

Other proteins that changed with Dox alone included Cdc42 effector protein 5 (Fig. 6E), which may be important given the role of Cdc42 in meiotic resumption and cell polarity (Cui et al, 2007). The most profound change observed was in diphthamide biosynthesis 2 (DPH2), which was reduced by Dox but restored by NMN co-treatment (Fig. 6K). Diphthamide is an evolutionarily conserved post-translational modification that occurs at a single histidine residue of translation elongation factor 2 (EF-2), which is needed for accurate protein translation by preventing -1 frameshift "slippage" during translation (Liu et al, 2012). Diphthamide biosynthesis genes are essential to normal development, and deletion of these genes in mice leads to developmental defects including embryonic lethality, while mutations found in humans are associated with developmental delays, intellectual disability, craniofacial and genital abnormalities (Nakajima et al, 2018). It is unclear why Dox treatment reduced ovarian DPH2 levels, nor why its levels were altered by NMN co-treatment. Further work should aim to investigate changes in EF-2 activity and its diphthamide modification in the context of chemotherapy induced infertility.

Importantly, any strategy to counteract the toxicity of chemotherapy must avoid compromising the ability of chemotherapy to reduce tumour growth. Given the ability of NMN to protect against chemotherapy induced toxicity in the ovary, one concern could be that systemic NMN treatment would have the effect of promoting tumour growth and/or reducing the oncological anti-tumour efficacy of chemotherapy, limiting its clinical utility. This question is especially pertinent given the role of NAD$^+$ in maintaining cancer cell metabolism (Navas et al, 2023; Tateishi et al, 2017), and the development of NAD$^+$ biosynthetic inhibitors targeting nicotinamide phosphoribosyltransferase (NAMPT) to treat cancer (Tateishi et al, 2017; von Heideman et al, 2010). We therefore sought to test whether NMN co-administration with the chemotherapy agents doxorubicin and cisplatin would impair the ability of these compounds to slow tumour growth in mice bearing an orthotopic xenograft of the highly aggressive, invasive and poorly differentiated triple-negative breast cancer cell line MDA-MB-231 (Fig. 7A–H).

This tumour experiment was conducted to test for the non-inferiority of NMN in combination with chemotherapy, versus

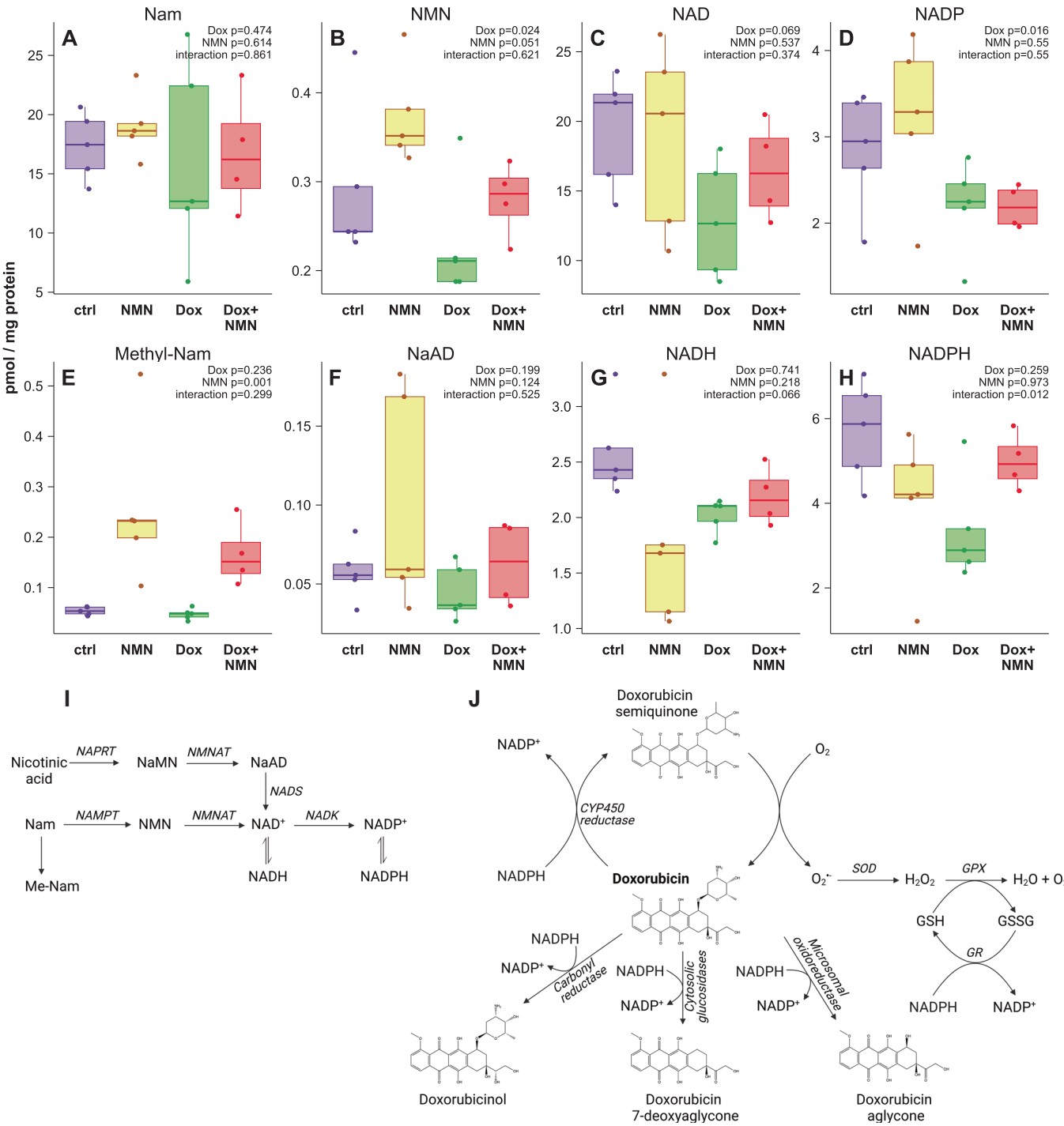

**Figure 5. Dox and NMN treatment impacts the ovarian NAD⁺ metabolome.**

Animals received an acute dose of doxorubicin (5 mg/kg) and/or NMN (200 mg/kg) as shown in Fig. 1A. Six hours later, animals were euthanased, ovaries collected and subject to mass spectrometry analysis for levels of (A) nicotinamide, (B) NMN, (C) NAD⁺, (D) NADP⁺, (E) 1-methyl-nicotinamide, (F) NaAD, (G) NADH and (H) NADPH. (I) Pathway for the incorporation of exogenous NMN treatment into the NAD⁺ metabolome, including into NADP⁺ and NADPH, which (J) is consumed during the metabolism of doxorubicin, which can generate reactive oxygen species that are neutralised by the glutathione (GSH) system, where the recycling of oxidised glutathione (GSSG) into GSH is also powered by NADPH. $n = 4$–5 per group, p-values between groups are from Bonferroni-adjusted t-tests derived from estimated marginal means analysis of a linear model of Dox and NMN treatment, with p-values for main and interaction effects of linear models annotated onto each panel. Tukey boxplots show the 25–75% interquartile range with whiskers indicating 95% confidence intervals, mean values indicated by a line within boxplots.

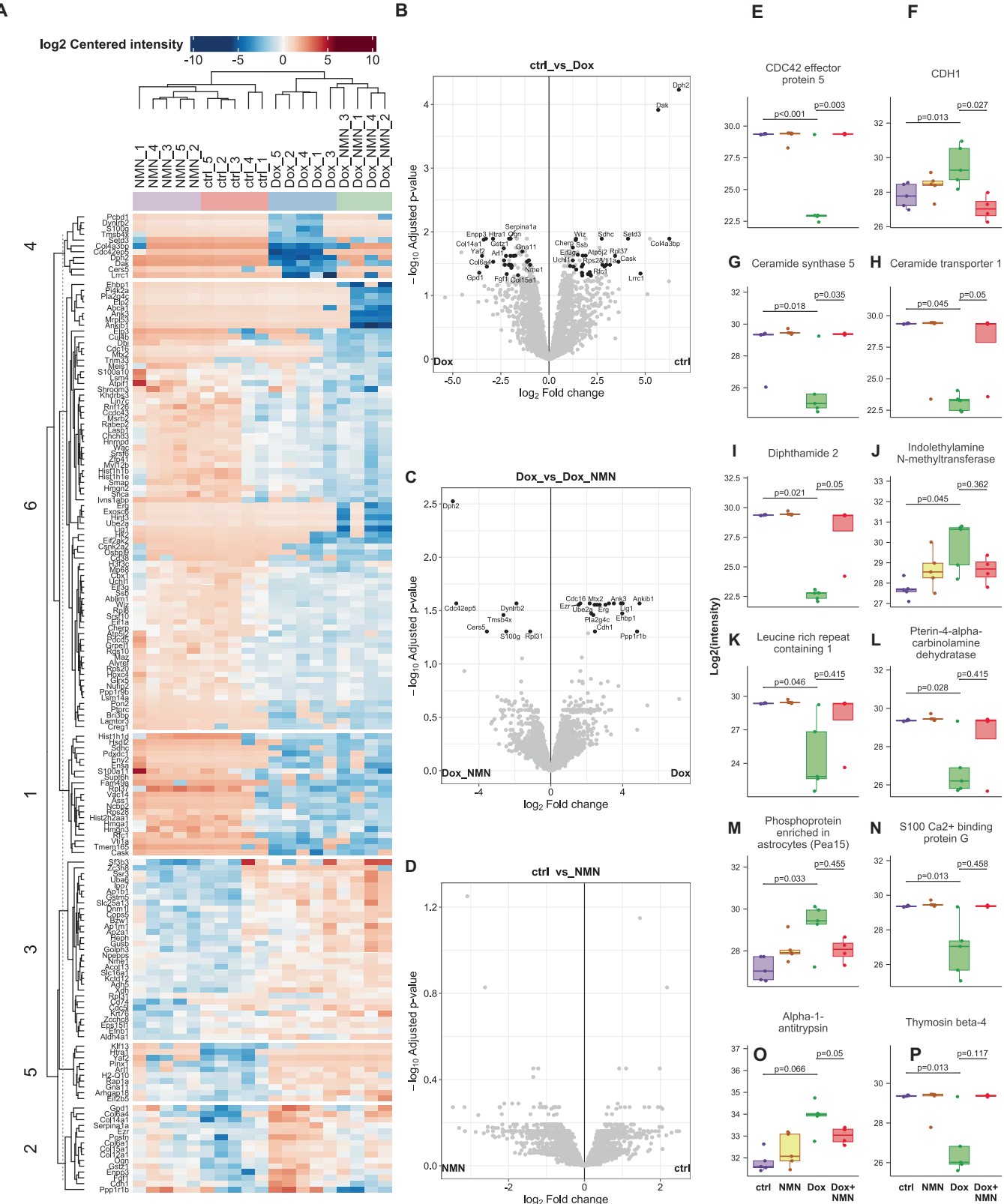

◀ **Figure 6. Proteomic analysis of ovaries following acute doxorubicin and NMN.**

As in Fig. 5, animals received an acute dose of doxorubicin and/or NMN, six hours later they were euthanased and ovaries rapidly collected for proteomics analysis using label-free whole cell proteomics. Significant differences in protein abundance are shown in (**A**) as clustered heatmaps, also presented (**B–D**) as volcano plots for select comparisons, showing $\log_2$ adjusted LFQ intensity. (**E–P**) Abundance levels for individual proteins that were significantly changed in samples that received Dox only compared to Dox + NMN and untreated controls, including (**E**) CDC effector protein 5, (**F**) CDH1, (**G**) ceramide synthase 5, (**H**) ceramide transporter 1, (**I**) dipthamide 2, (**J**) indolethylamine N-methyltransferase, (**K**) leucine-rich repeat containing 1, (**L**) pterin-4-α-carbinolamine dehydratase, (**M**) Pea15, (**N**) S100 $Ca^{2+}$ binding protein G, (**O**) α-1-antitrypsin, and (**P**) thymosin β-4. The criteria for significance was a minimum 2-fold difference in abundance between groups, with an adjusted $p$-value cut-off of 0.05. Statistics were performed using LFQ-Analyst, which creates linear models for each protein combined with Bayes statistics, and adjusted $p$-values based on the Benjamini–Hochberg method. $n = 4$–5 per group. Tukey boxplots show the 25–75% interquartile range with whiskers indicating 95% confidence intervals, mean values indicated by a line within boxplots.

chemotherapy alone, and to ensure a robust finding was repeated in two independent labs, with data from these separated into Fig. 7A–D and Fig. 7E–H. Surprisingly, treatment with NMN alone reduced tumour growth (Fig. 7A,E), while co-administration of NMN with both doxorubicin (Fig. 7B,F) and cisplatin (Fig. 7C,G) did not reduce their efficacy. To further test for non-inferiority of NMN co-treatment with chemotherapy against chemotherapy alone in a wider range of chemotherapy agents across entire dose-response curves in other cancer models, we next turned to in vitro studies in a further six human cancer cell lines, which were derived from endometrial (HEC-1A, RL95-2), mammary (MCF-7, MDA-MB-231) and ovarian (A2780 and SK-OV-3) cancers (Fig. 7I). These studies showed that NMN did not interfere with the dose–response curves of the chemotherapy agents bleomycin, cisplatin, doxorubicin, etoposide, gemcitabine, methotrexate, pacli-taxel, or vincristine, using in vitro cell viability (crystal violet staining) as a readout for their efficacy (Fig. 7I). The lack of effect of NMN on dose-response curves for chemotherapy further support the idea that NMN will not reduce the oncological efficacy of these drugs. Together, these in vivo and in vitro data suggest that an $NAD^+$ precursor such as NMN may be safe to use in the context of cancer, as a strategy to reduce the impact of cytotoxic chemother-apy on female fertility, without impairing cancer treatment.

## Discussion

Together, these data use orthogonal pharmacologic and genetic approaches to show that strategies to elevate $NAD^+$ can sustain ovarian function and fertility during chemotherapy. This has implications for the clinical management of female cancer patients of reproductive age or younger, including paediatric and adolescent cancer patients for whom ovarian stimulation and cryopreservation of oocytes or embryos prior to chemotherapy is not possible. These data are in line with previous findings around the role of $NAD^+$ precursors in female fertility during biological ageing (Bertoldo et al, 2020a; Habibalahi et al, 2022; Huang et al, 2022; Miao et al, 2020; Yang et al, 2020), as well as the impact of $NAD^+$ precursors in ameliorating the toxicity of chemotherapy towards other organ systems and in age-related disease (Li and Wu, 2021; Wu and Sinclair, 2016). These include impact of $NAD^+$ precursors on chemotherapy induced peripheral neuropathy (Hamity et al, 2020), hearing loss (Kim et al, 2014), and renal toxicity (Guan et al, 2017). This work adds another adverse impact of chemotherapy treatment that could potentially be ameliorated using $NAD^+$ precursors.

Interestingly, other work has investigated the impact of NMN treatment on infertility caused by chemotherapy and radiation

treatment, concluding there was no protection from NMN (Stringer et al, 2020). One explanation for this discrepancy is that $NAD^+$ precursors are likely to be most beneficial when $NAD^+$ homeostasis is challenged by insults including a need for drug detoxification, activation of the DNA damage response and altered metabolic flux, as occurs with chemotherapy treatment and/or ageing, but not in unchallenged scenarios where there is no excess demand on the $NAD^+$ metabolome. In the current study, NMN was provided concurrently with and following chemotherapy treatment, whereas the study by Stringer et al provided NMN to young, untreated animals for 7 days prior to irradiation or chemotherapy treatment, but not after (Stringer et al, 2020). When the $NAD^+$ metabolome is replete and undisturbed, as in young healthy animals that have not received chemotherapy, exogenous $NAD^+$ precursors are likely to be excreted as excess surplus, rather than accumulated. $NAD^+$ metabolites are excreted through the urine following their conversion to 1-methyl-nicotinamide, and the concept that excess material would be rapidly excreted matches our own data, where NMN treatment drastically increased the production of 1-methyl-nicotinamide (Fig. 5E). Other differences between these studies were the choice of ovarian insult, with our study using doxorubicin and cisplatin, rather than cyclophosphamide and γ-irradiation, and the choice of timepoint, with our studies conducted at 2 months post-chemotherapy, rather than 5 days. Interestingly, the trend towards increased numbers of corpora lutea during co-treatment with NMN and doxorubicin (Fig. 4G) was also mirrored during NMN treatment and cyclophosphamide and γ-irradiation (Stringer et al, 2020), though neither result was statistically significantly different.

This retention of ovulatory function (Fig. 1) and fertility (Figs. 2 and 3) after chemotherapy was not caused by NMN preventing a loss of primordial follicles (Fig. 4A), but rather by reducing atresia in primordial and preantral follicles. We hypothesize that Dox+NMN treatment had both short- and long-term effects on folliculogenesis and fertility. In the wave of folliculogenesis occurring at the time of chemotherapy, co-treatment with NMN attenuated follicle atresia caused by Dox, leading to the survival of growing antral follicles as evidenced by the increased numbers of large antral follicles (Fig. 4F) containing viable COCs (Fig. 1B,E) which had the capacity to support pregnancies and development to term (Figs. 2 and 3). However, the Dox+NMN treatment also led to more pups than Dox alone in the subsequent waves of folliculogenesis (i.e. from 50 to 250 days post chemotherapy), which cannot be attributed to changes observed in the growing follicle population 8 weeks after treatment (Fig. 4). This prolonged benefit of NMN on fertility may have been caused by the protection of toxic effects of Dox on ovarian reserve

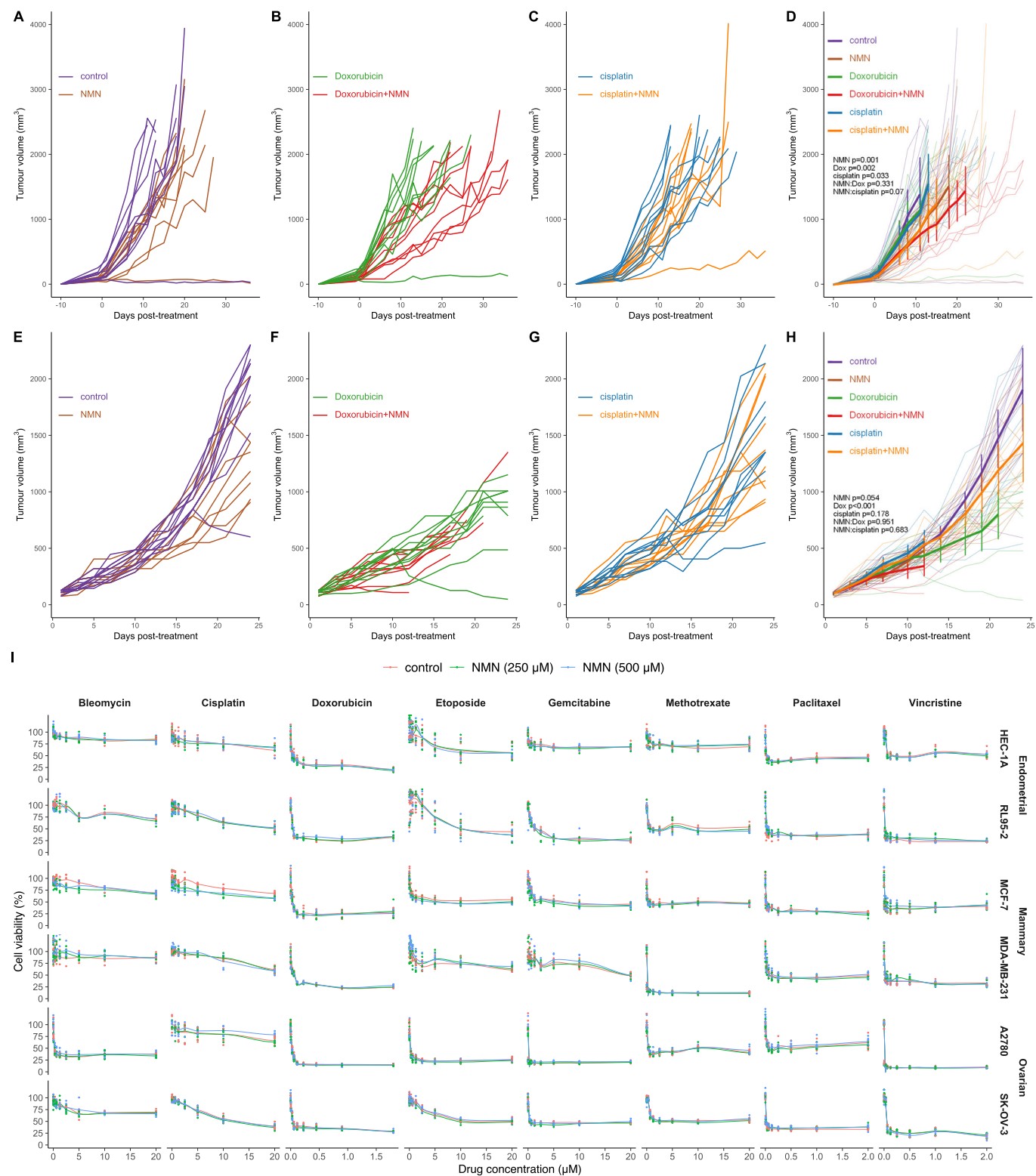

health (Fig. 4A), or by the ongoing administration of NMN alone during this period affecting folliculogenesis and ovulation rate independent of Dox treatment (as per Fig. 2G), or alternatively NMN detoxifying effects of Dox on non-ovarian tissues or organs such as the uterus (Griffiths et al, 2020).

Interestingly, the impacts of NMN treatment to fertility were not restricted to delivery at the time of chemotherapy. We used a previously described model of cisplatin induced infertility (Gonfloni et al, 2009; Kerr et al, 2012), whereby cisplatin is delivered to prepubertal mice (Fig. 3A), when the primordial reserve is highest. As

**Figure 7. NMN does not reduce the efficacy of chemotherapy.**

The MDA-MB-231 orthotopic xenograft model of mammary cancer was used in mice treated with vehicle, doxorubicin or cisplatin in the presence or absence of NMN. Data are separated into the effects of NMN with (A) vehicle, (B) doxorubicin and (C) cisplatin, summarised in (D) with thicker lines indicating mean tumour volumes up to the point of the first animal being euthanased from each cohort, and error bars as 95% confidence intervals. (E–H) The same experiment was repeated independently in a second laboratory. $n = 10$ per group for both experiments, error bars are mean ± 95% CI. Data analysed by Baysesian joint modelling of tumour growth and survival, incorporating survival data and mixed linear modelling of tumour growth over time to account for the removal of animals due to ethical limits to maximum tumour volume. This joint modelling was used to test for main effects and interactions of NMN and chemotherapy on tumour growth as a longitudinal factor, p-values as indicated. (I) To further test for the non-inferiority of NMN co-treatment with chemotherapy, NMN (250 and 500 μM) was also tested in vitro for potential interactions with bleomycin, cisplatin, doxorubicin, etoposide, gemcitabine, methotrexate, paclitaxel and vincristine, in the endometrial, mammary and ovarian cancer cell lines HEC-1A, RL95-2, MCF-7, MDA-MB-231, A2780 and SK-OV-3, using crystal violet staining as an indicator of cell viability.

NMN is delivered in the drinking water, treatment was not possible in lactating animals, and these animals only started NMN treatment at weaning, 2 weeks after the cisplatin insult. Despite NMN treatment not starting until 2 weeks after cisplatin, we still observed improved fertility (Fig. 3), suggesting that the benefits of NMN may be related to ongoing repair processes in the ovarian reserve. The mechanism for this finding should be subject to follow-up study.

The difference in fertility (Figs. 2 and 3) and follicle health with Dox+NMN treatment may be related to supporting levels of NADPH (Fig. 5H), which is required to regenerate the cellular antioxidant glutathione in the face of ROS generation from Dox treatment. While these measurements were taken in the context of Dox treatment, it is worth noting that the same strategies also protected against decreased oocyte yields (Fig. 1E,F) and fertility (Fig. 3) caused by treatment with cisplatin. As with doxorubicin, cisplatin treatment also results in ROS generation, which is neutralised by the NADPH-powered glutathione system, warranting investigation of this strategy for other forms of chemotherapy which are detoxified via this system. Together, this rescue in follicle health may provide a physiological mechanism by which $NAD(P)^+$ repletion protects against the fertility destroying effects of chemotherapy. In this scenario, the flux of metabolic precursors through to NADPH formation could be rate limiting to follicle health during chemotherapy. One question is whether the availability of $NAD^+$ precursors or the expression and activity of $NAD^+$ biosynthetic enzymes is more important to this flux. Overexpression of the NAD biosynthetic enzymes NMNAT1 and NMNAT3 provided some protection against a loss in COC yield (Fig. 1C,D,F), however, the effect size was less robust than for NMN treatment (Fig. 1B,E). The direct substrate for NMNAT1 and NMNAT3 is NMN, which was decreased by Dox treatment (Fig. 1B), and it may be that declining substrate availability is rate-limiting to $NAD(P)^+$ flux, providing a mechanism for the impact of NMN treatment on ovarian function and fertility. One other possibility for altered $NAD^+$ flux during chemotherapy treatment could be altered expression of the $NAD^+$ consuming enzyme CD38 (Perrone et al, 2023; Yang et al, 2024), the increased expression of which is a cause of doxorubicin induced cardiotoxicity (Peclat et al, 2024), however, our proteomics data (Fig. 6) did not show any change in the levels of this enzyme with Dox treatment. Another potential mechanism could be the role of $NAD^+$ consuming PARP enzymes which are involved in repairing DNA damage, which could be induced by chemotherapy. Our in vitro experiments with PARP inhibitors were inconclusive, and future work should aim to investigate this in more detail.

Although our observations would be consistent with a model whereby repletion of the $NAD(P)^+$ metabolome could provide a

buffer to replenish NADPH levels in the ovary, further work will be needed to determine the extent to which these systems are relevant to the protection against ovarian dysfunction and infertility. The metabolism of xenobiotics such as doxorubicin primarily occurs in the liver, however we readily detected the expression of NADPH-dependent enzymes that carry out doxorubicin metabolism in the proteomes of these ovary samples (Fig. 6). Although we cannot use raw peptide intensity values to compare the abundance of different protein species, it is notable that we could readily detect high numbers of peptides for NADPH-dependent carbonyl reductase 1 (CBR1) (supplementary files). This is consistent with observations that CBR1 is highly abundant in the ovary, composing 1–4% of all cytosolic protein (Iwata et al, 1990b), where its activity fluctuates during the ovarian cycle (Espey et al, 2000; Iwata et al, 1990a). The activity of this enzyme in the ovary could be a double-edged sword: on the one hand, the NADPH-dependent metabolism of DOX into doxorubicinol by CBR1 allows for the subsequent conjugation and excretion of this drug, and in doing so could deplete NADPH levels needed for the regeneration of GSH needed to neutralise ROS. On the other hand, doxorubicinol is the metabolite of Dox responsible for cardiomyopathy, though this occurs through its interaction with $Ca^{2+}$ channels (Olson et al, 1988) which may be less relevant to ovarian function. Further, we found that NMN co-treatment with Dox did not increase cardiac dysfunction, which is in line with other recent work testing the impact of administering $NAD^+$ precursors, or inhibiting the $NAD^+$ consuming enzyme CD38 (Margier et al, 2022; Peclat et al, 2024; Zheng et al, 2019). Future work should aim to test this concept by comparing the levels of chemotherapy metabolites in ovaries from animals treated with or without NMN. Finally, it will be important to expand this investigation to a broader list of chemotherapy agents and to avoid extrapolating these findings beyond doxorubicin, the agent which was primarily used in these studies. While cytotoxic chemotherapy agents remain a mainstay of oncology, it is likely that profoundly different mechanisms of action will limit the utility of $NAD^+$ boosting strategies to protect against infertility.

The observation of reduced tumour growth with NMN alone (Fig. 7) was surprising, as this experiment was originally intended to measure non-inferiority of NMN co-treatment compared to chemotherapy alone. Further work is required to understand how this occurred. It is likely that the tumour response to elevated $NAD^+$ is likely to vary by mutation, with caution warranted. As these xenograft studies were in immunocompromised animals, enhanced immune surveillance is an unlikely mechanism. One hypothesis is that increasing $NAD^+$ availability impacts the metabolism of cancer cells, resulting in a shift away from Warburg-type glycolytic metabolism towards oxidative

phosphorylation, as we proposed previously (Wu et al, 2014). Although in vivo treatment suggested that NMN could have an impact on tumour growth, in vitro experiments with a matrix of chemotherapy and cancer cell lines showed no impact of NMN on cell survival—suggesting that any impact may be due to altered growth kinetics, rather than interfering with the efficacy of chemotherapy. Regardless, these non-inferiority data provide encouraging early evidence that $NAD^+$ raising therapies may be safe to use in cancer patients, though the heterogeneity of tumour types means that these studies should be interpreted with caution, and additional studies are needed in different cancer models. Our observation of enhanced chemotherapy action towards tumours due to NMN co-treatment is in line with previous findings for nicotinamide riboside (NR) and paclitaxel treatment (Hamity et al, 2020). More recently, Jiang et al also showed that NMN treatment could slow tumour growth and metastases in a similar triple-negative breast cancer orthotopic xenograft tumour model (Jiang et al, 2023), again supporting our findings (Fig. 7).

There were several limitations of our study. As a model for chemotherapy induced infertility we used two chemotherapy agents (doxorubicin and cisplatin), testing three interventions (NMN treatment, NMNAT1-Tg and NMNAT3-Tg overexpression) and using three outcomes for fertility (COC yield, breeding, ovarian histology). It would have been ideal to test every permutation of these insults, interventions and outcomes, however, we were unfortunately limited by animal availability and cost. These experiments could be conducted in future work, along with testing a wider range of chemotherapy interventions, including clinically relevant chemotherapy combinations. Future work should also aim to test a wider range of chemotherapy doses: it is likely that this intervention will shift the dose–response curve of ovarian toxicity, and while our work showed a promising reduction in infertility against previously described gonadotoxic doses (Ben-Aharon et al, 2010; Gonfloni et al, 2009; Kerr et al, 2012; Nguyen et al, 2019), it is unclear whether this protection will translate to higher doses. Another aspect of these experiments that requires further work is in the timing of treatment: we exposed animals to NMN before, during, and long after the chemotherapy insult. While the metabolomics (Fig. 5) and proteomics (Fig. 6) data may provide interesting hints as to potential mechanism, these samples were taken at an early timepoint only 5 h after Dox treatment. This period of treatment might not be relevant, and it may be the case that the protection offered by NMN is a long-term, chronic effect that takes place over weeks of treatment. If so, those molecular changes (Figs. 5 and 6) may be irrelevant to the mechanism of action for NMN treatment, and future work should aim to narrow the window of when NMN treatment offers its protection, to help identify when these molecular changes might be relevant to this phenotype. Metabolomics data can have a wide data distribution, and the experiments in Fig. 5 are likely under-powered: once the timeframe of efficacy for NMN treatment can be established, these experiments could be re-examined using higher experimental power. An important aspect for future work that we highlighted was that $NAD^+$ consuming PARP enzymes could be triggered by chemotherapy in the ovary, leading to an $NAD^+$ decline. While there was a trend towards declining $NAD^+$ levels with chemotherapy (Fig. 5C), future experiments should aim to clearly test this possibility, ideally through in vitro treatment with small molecule PARP inhibitors. To better elucidate a potential mechanism that

would explain our findings, single-cell RNA sequencing should be considered for future work. To complement this, in situ immuno-histochemistry staining for DNA damage and apoptosis markers in ovaries could, at a minimum, provide hints around which cell types are likely to contribute to the observed phenotype. We attempted these studies, but observed considerable variability in staining patterns and intensity within treatment groups, and no treatment effects—as described above, this may relate to an inappropriate selection of timepoints, or other important parameters. Our inability to precisely identify both the molecular mechanism and the relevant cell type(s) that mediate this phenotype are a weakness of this study, and future work should aim to address this in greater detail using in situ immunohistochemistry and single-cell sequencing.

These studies reveal that $NAD^+$-raising compounds represent an effective pharmacological approach to reduce female infertility caused by cancer treatment. This has immediate implications for the clinical management of cancer patients by alleviating the risk of ovarian failure from chemotherapy as a consideration in oncology, providing a potential alternative to cryopreservation protocols that require invasive superovulation and oocyte pick-up procedures, which can delay cancer treatment, and may have limited success rates (Bertoldo et al, 2020b; Woodruff, 2009). Here, we demonstrated that co-treatment with NMN during either doxorubicin or cisplatin treatment did not compromise the efficacy of these agents, and in the case of the MDA-MB-231 orthotopic xenograft, surprisingly, reduced tumour growth. This work represents a clinically tractable intervention to non-invasively preserve ovarian function and female fertility during chemotherapy without diminishing the efficacy of cancer treatment, potentially acting as a fertoprotective neoadjuvant agent (Woodruff, 2017) to improve the long-term health and quality of life of cancer survivors.

## Methods

**Reagents and tools table**

| Reagent/Resource | Reference or Source | Identifier or Catalog Number |
| --- | --- | --- |
| **Experimental Models** | | |
| C57BL/6JAusb (*Mus musculus*) | Australian BioResources, Moss Vale, NSW Australia | C57BL/6JAusb |
| NMNAT1 transgenic mice | (Yahata et al, 2009). Obtained under material transfer agreement from NCPP, Japan | B6.Cg-Tg(CAG-Nmnat1)1Ara |
| NMNAT3 transgenic mice | | B6.Cg-Tg(CAG-Nmnat31Ara |
| MDA-MB-231 cell line | American Type Culture Collection (ATCC) | MDA-MB-231 |
| RL95-2 cell line | | RL95-2 |
| MCF-7 cell line | CellBank Australia | 86012803 |
| HEC-1-A cell line | | JCRB1117 |
| A2780 cell line | | 93112519 |
| SK-OV-3 cell line | | 91091004 |
| **Chemicals, Enzymes and other reagents** | | |
| Doxorubicin hydrochloride | Cayman Chemicals, USA | 15007 |
| Cisplatin | Enzo, Switzerland | ALX-400-040 |

| Reagent/Resource | Reference or Source | Identifier or Catalog Number |
|---|---|---|
| Nicotinamide mononucleotide (NMN) | GeneHarbor Biotechnology, Hong Kong | |
| Paraffin oil | EMD Millipore, USA | ES-005-C |
| Periodic acid | POCD Scientific, Artarmon, NSW, Australia | |
| Scott's blue | | |
| Schiff's reagent | Thermo Fisher Scientific, Scoresby, VIC, Australia | J62171.AP |
| Mayer's hematoxylin | Sigma-Aldrich, Castle Hill, NSW, Australia | MHS32 |
| DPX mounting media | | 06522 |
| Crystal violet | | C0775 |
| M2 medium | | M7167 |
| 3-isobutyl-1-methylxanthine (IBMX) | | I7018 |
| D,L-Dithiothreitol (DTT) | | D9779 |
| M16 medium | | M7292 |
| Matrigel | Corning | |
| Bleomycin | National Cancer Institute, USA | NCI 10 mM approved oncology drugs set IV |
| Etoposide | | |
| Gemcitabine | | |
| Methotrexate | | |
| Paclitaxel | | |
| Vincristine | | |
| Schiff's reagent | Thermo Fisher Scientific, Scoresby, VIC, Australia | 3952016 |
| Pierce™ BCA Protein Assay Kit | | 23225 |
| TPCK-Trypsin | | 20233 |
| Veet hair removal cream | Reckitt Benckiser, NSW, Australia | |
| Pregnant mare's serum gonadotrophin (Folligon) | Intervet, Boxmeed, Netherlands via MSD Animal Health, Australia | |
| **Software** | | |
| MBF Stereo Investigator | MBF Bioscience, USA | v2021.1.1 |
| R | | v4.4.1 |
| Rstudio | | v2024.04.1 |
| emmeans (Estimated Marginal Means, aka Least-Squares Means) | https://CRAN.R-project.org/package=emmeans | v1.8.0 |
| Jmbayes2: Extended Joint Models for Longitudinal and Time-to-Event Data | https://cran.r-project.org/web/packages/JMbayes/index.html | v0.3-1 |
| ggplot2: | https://ggplot2.tidyverse.org | v3.4.0 |
| ggtext | https://cran.r-project.org/web/packages/ggtext/index.html | v0.1.2 |
| ggpubr | https://cran.r-project.org/web/packages/ggpubr/index.html | v0.4.0 |
| ggsignif | https://doi.org/10.31234/osf.io/7awm6 | v0.6.3 |

| Reagent/Resource | Reference or Source | Identifier or Catalog Number |
|---|---|---|
| ggh4x | https://cran.rstudio.com/web/packages/ggh4x/index.html | v0.2.3.9000 |
| tidyverse | https://doi.org/10.21105/joss.01686 | v2.0.0 |
| dplyr | https://github.com/tidyverse/dplyr | v1.1.4 |
| Chemdraw | Perkin Elmer | v19.0 |
| MaxQuant | (Cox and Mann, 2008) | v1.6.17 |
| LFQ-Analyst package v1.2.3 | (Shah et al, 2020) | v1.2.3 |
| BioRender | | Biorender.com |
| Xcalibur | ThermoFisher Scientific, Waltham, MA, USA | v2.2 |
| **Other** | | |
| ViusalSonics Vevo 2100 Ultrasound | Fujifilm VisualSonics, Japan | |
| Technovit 7100 (resin embedding) | Heraeus Kulzer, Germany | |
| Leica RM2252 microtome | Leica Biosystems, USA | |
| Olympus VS200 Research Slide Scanner | Olympus Life Science, Japan | |
| UPLXAPO 40x NA 0.95 objective | | |
| SpeedVac Plus | Savant | SC110A |
| Ultimate nanoRSLC UPLC and autosampler system | Dionex, Amsterdam, Netherlands | |
| Micro C18 precolumn | | |
| Valco 10 port UPLC valve | Valco, Houston, TX | |
| C18AQ media, 120 Å | Dr Maisch, Ammerbuch-Entringen Germany | |
| Orbitrap Fusion Lumos | Thermo Electron, Bremen, Germany | |
| TSQ Vantage | ThermoFisher Scientific, Waltham, MA, USA | |
| Menzel-Glaser Superfrost™ Plus | Thermo Fisher Scientific, Sydney, NSW, Australia | |
| KNITTEL Coverglass | ProSciTech, Kirwan, Queensland, Australia | |

## Animals

### Ethics

Experiments at UNSW were carried out with prior approval of the UNSW Animal Care and Ethics Committee (ACEC) under ACEC numbers 13/134B, 15/134A, 18/133A and 19/24B. UNSW ACEC operates under animal ethics guidelines from the National Health and Medical Research Council (NHMRC) of Australia. Other experiments were carried out as contract research at Washington Biotechnology Inc (Baltimore, MD, USA) and Charles River Laboratories (Worcester, MA, USA) under the Office for Laboratory Animal Welfare (OLAW) numbers A4192-01 and A4645-01, respectively. Both facilities are accredited by the Association for Assessment and Accreditation of Laboratory Care (AAALAC).

### Housing

For experiments at UNSW, transgenic and wild-type strains were bred at Australian Bio-Resources (ABR) in Moss Vale, NSW Australia and delivered to UNSW Sydney at 6–7 weeks of age. All animals were on the C57BL6 background. Animals were maintained in individually ventilated cages at 22 °C at 80% humidity at a density of up to 5 per cage, with ad libitum access to food and water. All water in this animal house was acidified to pH 3 with HCl to decrease microbial growth. Animals were maintained on standard chow diet obtained from Gordon's Specialty Stock Feeds (Yanderra, NSW Australia) as described previously, briefly, this diet contained 8% calories from fat, 21% from protein, and 71% from carbohydrates, with an energy density of 2.6 kcal/g. The UNSW animal house maintained a 12 h light/dark cycle with lights on at 0700 and off at 1900. The NMNAT1 and NMNAT3 transgenic mouse strains used here were as previously described (Yahata et al, 2009), and the colony was maintained through breeding heterozygous males with wild-type females, with a subsequent Mendelian ratio of offspring to be used for experiments of 50% heterozygous transgenics, and 50% wild-type littermate controls. Following delivery from the breeding centre at ABR Moss Vale, animals were acclimatised at the UNSW Biological Resource Centre (BRC) for at least 1 week prior to experiments.

### NMN and chemotherapy treatments

For NMN treatment, mice were maintained on standard acidified drinking water in the presence or absence of NMN (Geneharbor Hong Kong Technologies Ltd, Hong Kong) at 2 g/L, with water bottles changed twice weekly to prevent microbial growth, with treatment maintained until the end of the experiment.

(1) Treatment with NMN and chemotherapy commenced at 8 weeks of age in female mice. In these experiments, NMN treatment started 3 days prior to a single intraperitoneal injection of either doxorubicin or cisplatin.
   a. Doxorubicin hydrochloride (10 mg/kg, Cayman Chemicals, USA) was dissolved initially in DMSO and then diluted 1:20 in saline for a final DMSO concentration of 5% in the injection.
      i. Doxorubicin was injected by an investigator not involved in subsequent experiments, due to the bright red colour of doxorubicin that could unblind the investigator to treatment groups.
   b. Cisplatin (5 mg/kg, Enzo, Switzerland) was dissolved in saline, with or without a co-injection of NMN (200 mg/kg) dissolved in saline.
      i. Cisplatin was at no point exposed to DMSO, which can alter the chemical coordination of the Pt atom in cisplatin (Hall et al, 2014). Dose volumes were 100 μL per 20 g body weight
(2) Eight weeks after chemotherapy treatment, animals were either stimulated with PMSG and sacrificed to obtain oocyte yield (see "Ovarian stimulation and oocyte collection" protocol below), or sacrificed without prior PMSG treatment for ovarian histology (see "Follicle health and quantification" protocol below), or introduced to a male of proven breeding ability for timed breeding trials (see "Breeding trial" protocol below).
(3) In experiments involving NMNAT transgenic strains, eight animals

received a single injection of either doxorubicin (10 mg/kg) or cisplatin (5 mg/kg) at 7–9 weeks of age as above but did not receive NMN. Comparisons were between transgenic mice and their wild-type (WT) littermates.
(4) For all experiments, body weights and monitoring were recorded on a weekly basis and are summarised in Appendix Fig. S2.

### Ovarian stimulation and oocyte collection

(1) For oocyte collection studies (Fig. 1), animals received a single intraperitoneal injection of 7.5 IU Pregnant mare's serum gonadotropin (PMSG) (MSD Animal House, Australia).
(2) Between 44–46 h later, animals were euthanased by cervical dislocation, and ovaries rapidly collected.
(3) Fat and tissue surrounded the ovaries were carefully and rapidly removed under a dissecting microscope.
(4) Oocytes were mechanically released from ovaries under M2 medium supplemented with 3 mg/ml bovine serum albumin (BSA) in the presence of the meiotic arrest inhibitor 3-isobutyl-1-methylxanthine (IBMX) (Sigma-Aldrich, Australia) at 50 μM.
(5) The released oocytes, including cumulus oocyte complexes (COC) and denuded oocytes (DO) in the germinal vehicle (GV) stage were collected by aspirator tubes (Sigma-Aldrich), and transferred into another drop of IBMX-M2 media overlayed with paraffin oil (EMD Millipore Corporation, USA), which was pre-warmed to 37 °C.
(6) Oocytes were then counted to obtain data shown in Fig. 1, and the cumulus cells removed by mechanical disruption with a mouth pipette.
(7) Oocytes were then rinsed 3 times in IBMX-free M2 (Sigma-Aldrich, Australia) containing 3 mg/ml BSA, which had been equilibrated for 24 h in 5% $CO_2$, to clear off the IBMX from medium, then transferred into another clean, pH calibrated M16 medium that was overlayed with oil and returned to the incubator under 37 °C in a humidified atmosphere of 5% $CO_2$ to assess subsequent meiotic development.
(8) Meiotic maturation measurements included the rates of GVBD and first polar body extrusion (PBE). The rates of GVBD were measured by counting the number of oocytes where there was a disappearance of nuclear membranes in the first two hours after release. Oocytes that failed to develop into the GVBD stage after 2 h were discarded, and the remaining GVBD stage oocytes were then continuously incubated to assess PBE development, indicated by the production of first polar body, in 14 h, 16 h and 20 h afterward.

### Breeding trial

Breeding trials started at 16 weeks of age. Animals were subject to a timed-mating protocol, whereby prior exposure of females to dirty bedding from males allowed for stimulation and synchronization of the females' estrous cycles through the Whitten Effect (Jemiolo et al, 1986). Breeding trials in Fig. 2 followed doxorubicin treatment at 8 weeks of age as described in the "NMN and chemotherapy treatment" protocol above, while for the breeding trial in cisplatin-treated mice (Fig. 3), we used a previously described paradigm or

chemotherapy induced ovarian depletion, with cisplatin (2 mg/kg) or saline control injections into 7-day-old mice, in the absence of NMN treatment. These animals were weaned as per normal at 3 weeks of age onto normal drinking water or water containing NMN (2 g/L). Doses for doxorubicin and cisplatin (CDDP) doses were based on previous models for chemotherapy induced infertility (Ben-Aharon et al, 2010; Kerr et al, 2012). Body weights were recorded on a weekly basis and are summarised in Appendix Fig. S2.

(1) Each breeding round commenced with group-housed female mice being 'scented' using dirty bedding from a male's cage for 3 days.
(2) Each female was then placed into the cage of an individually housed male of proven fertility in a 1:1 ratio ('co-habitation'). This transfer was made in the evening, in anticipation of night-time ovulation.
(3) The following early morning each female was assessed for the presence of a vaginal plug as evidence of copulation and, regardless of plug status, were separated to prevent daytime mating.
   a. Females with no plug were replaced with the same male the following evening to undergo another night of mating.
   b. If, following 3 consecutive nights of co-habitation, no plug had been observed, females were separated and observed for weight gain for 5 days to account for possibility of missed plug, before undergoing re-scenting.
   c. A female would repeat the current mating round until a positive plug was confirmed.
   d. Females that had copulated, as demonstrated by plug presence, but did not fall pregnant would be deemed to have failed that mating round and would commence the next mating round with a different male stud, and so on.
(4) Females with a positive plug were considered to have passed that mating round and would be housed separately.
(5) At day 15 post-copulation (pc), mice would undergo micro-ultrasound (VisualSonics Vevo 2100 Ultrasound) to determine pregnancy state, as determined by the presence of a foetal heartbeat.
(6) Offspring were maintained with females until weaning at 21 days.
(7) After weaning, females were returned to additional mating rounds, for a total of 6 mating rounds per animal.

The primary endpoint for this study was the number of live offspring per round of confirmed mating, which was determined by the presence of a vaginal plug as described above, and the total number of live offspring for each cohort, with $n = 10$ animals per group.

## Follicle health and quantification

To assess ovarian reserves, ovaries were collected from diestrus stage animals, embedded in resin, subject to thick (20 µm) sections for the entire ovary, with every third section digitally scanned for stereology analysis, and follicle reserve and health manually counted on a grid using Stereo Investigator software. Slides were labelled with a code that was separately assigned to each sample by an independent investigator, so that all analyses were conducted under blinded conditions. Detailed descriptions of each step of this analysis are provided below.

(1) *Estrous cycle tracking*: estrous cycles were tracked for 5 consecutive days to determine when mice were at diestrus, to allow for ovaries to be collected at a consistent stage of their cycle.
   a. Each morning, mice were removed from their cages and placed onto the wire rack lids.
   b. A yellow-tip 20 µL pipette was used to gently expel 20 µL of 0.9% saline into the vagina, and immediately aspirated back into the pipette.
   c. This aspirate was pipetted onto glass slides, and allowed to dry at room temperature.
   d. Slides were then dipped into a filtered solution containing 0.5% Toluidine blue in 20% ethanol.
   e. Slides were then dipped into water, and examined under a light microscope.
   f. The presence and number of epithelial cells—cornified or nucleated, and leukocytes was assessed to determine the cycle stage, defined as: estrous—mostly cornified epithelial cells; metestrus—both cornified epithelial cells and leukocytes; diestrus—mostly leukocytes; proestrus—mainly nucleated, some cornified epithelial cells.
(2) *Tissue collection*: when animals were confirmed to be at diestrus, following vaginal smearing as per above, they were euthanased for tissue collection as follows:
   a. Mice were placed in an isofluorane anaesthesia chamber, set to 3–5% isofluorane.
   b. Once immobile, individual animals were removed from the chamber and placed on the bench with a nose cone that continued to deliver isofluorane.
   c. Anaesthesia was then confirmed using the toe pinch test, to ensure that animals did not physically react after the toe being pinched.
   d. Blood was collected by cardiac puncture using a 1 ml tuberculin syringe and 30-gauge needle (BD Medical, USA).
      i. Blood was then transferred to a micro centrifuge tube and left at room temperature to clot for 30 min.
      ii. Centrifugation was then performed at $5000 \times g$ for 10 min. to collect serum, which was subsequently transferred to a new tube and stored at $-80\,°C$.
   e. After cardiac puncture, mice were immediately euthanized by cervical dislocation, and tissues rapidly collected.
   f. Ovaries were carefully cleaned from remaining fat under a dissecting microscope and weighed.
   g. One ovary from each animal was fixed in 4% paraformaldehyde (PFA) at $4\,°C$ overnight and transferred to 70% ethanol the next day for storage.
      i. The other ovary was placed in a micro centrifuge tube and snap frozen in liquid nitrogen for molecular analyses.
(3) *Resin embedding*: Each slide contained at least five serial sections, and was marked with a code that did not indicate treatments, allowing assessment to be performed in a blinded fashion.
   a. Fixed ovaries stored in ethanol were dehydrated and embedded in resin with a kit by Technovit 7100 (Heraeus Kulzer, Wehrheim, Germany) according to the protocol provided by the manufacturer.
   b. The resin blocks containing the fixed ovarian samples were allowed to solidify and dry before they were taken out of the mould at 3 weeks.

c. Resin blocks were then sectioned at 20 μm on a Leica RM2252 microtome with a 16 cm d-profile TC knife.

d. The cut sections were immersed in RO water, unfolded, straightened and placed on glass slides (Menzel-Glaser Superfrost™ Plus, Thermo Fisher Scientific, Sydney, NSW, Australia).

e. All slides were placed on a heat plate at 37 °C to dry and set and subsequently stored at room temperature.

(4) *Schiff staining of resin sections*

a. Dried ovarian sections were submerged in a bath of periodic acid (POCD Scientific, Artarmon, NSW, Australia) for 30 min.

b. Slides were then rinsed with tap water for 5 min.

c. Slides were then stained with Schiff's reagent (Thermo Fisher Scientific, Scoresby, VIC, Australia) for 45 min at room temperature in a fume hood.

d. Slides were rinsed with tap water for 5 min.

e. Slides were counterstained with filtered Mayer's hematoxylin (Sigma-Aldrich, Castle Hill, NSW, Australia) for 2 h in an incubator at 37 °C.

f. Slides were rinsed again with tap water for 5 min.

g. Slides were then incubated in Scott's blue (POCD Scientific, Artarmon, NSW, Australia) in the fume hood for 3 min at room temperature.

h. Slides were washed in tap water for 3 min and left to dry overnight.

i. The following day, DPX mounting media for histology (Sigma-Aldrich, Castle Hill, NSW, Australia) was dropped onto each section prior to mounting with coverslips (KNITTEL Coverglass, ProSciTech, Kirwan, Queensland, Australia).

(5) *Slide scanning*: The Olympus VS200 Research Slide Scanner was used to acquire datasets for stereology analysis. Overview images of the whole slides were captured with the PLAN 2x NA 0.06 objective. The UPLXAPO 40x NA 0.95 objective was used to acquire a detailed virtual z-stack with 1 μm increments of each section with 1 ms exposure. The captured 40x virtual z-stack was then loaded into MBF Stereo Investigator (MBF Bioscience, USA) for stereology analysis.

(6) *Stereology*: Stereo Investigator (SI) software was used to obtain accurate ovarian follicle counts, as previously described (Sarma et al, 2020). All histology sample preparation, slide scanning, follicle counting and health assessments were performed in a blinded fashion, with samples labelled with codes that were unknown to the investigator analysing each sample.

a. Scanned sections were imported as a virtual z-stack in SI with a cut thickness of 20 μm and mounted thickness of 15 μm.

b. Every third consecutive section of each ovary was assessed by a blinded investigator.

c. Lens calibration was performed to match the image resolution. Large growing follicles, including secondary, small antral, large antral, and pre-ovulatory follicles were counted across the entire section. They were only included in the count if the nucleus of the oocyte was visible to prevent overcounting.

d. Corpora lutea (CLs) were counted per ovary, while tracking across sections to confirm a CL.

e. According to the workflow, the region of interest (all ovarian tissue) was selected under low magnification with the auto-trace assist.

f. The grid size was then set to 800 and counting frame to 500.

g. All follicles were manually counted within the square counting frame.

h. The frame had two of its sides coloured red and two coloured green. For follicles lying on a line, only follicles lying on green lines were included in the count, while follicles crossed by the red line were excluded.

i. Follicles were counted from every third section of the entire ovary, and used by Stereo Investigator software to calculate the follicle population estimate for the entire ovary. This calculation accounts for area sampling fraction (counting frame/grid size), section sampling fraction (every third section), and mounted thickness over counting frame height, as per the software default. Unhealthy follicles were also assessed according to their size—large follicles across the entire section and small follicles on the grid.

j. Data were recorded with codes only, and unblinding to experimental groups for each sample only occurred once all data had been recorded.

(7) *Follicle classification*: Morphological classification of the ovarian follicles was performed according to defined standards in the field, e.g. (Bertoldo et al, 2021; Pedersen and Peters, 1968).

a. Primordial follicles were described by an oocyte surrounded by a single layer of squamous granulosa cells (GCs).

b. The transition stage between primordial and primary follicle is defined by an oocyte and a single layer of GCs, some of which can be squamous and some cuboidal.

c. Primary follicles consist of an oocyte and a single layer of cuboidal GCs.

d. Primordial follicles are described as "non-growing" or the "ovarian reserve", and transitory and primary as "small growing" follicles.

e. Secondary follicles are described as having a growing oocyte and at least two complete layers of cuboidal GCs.

f. Antral follicles are defined by two or more GC layers and an antral space. Depending on the volume of the antral space, the antral follicle can be classified as small antral or large antral. A pre-ovulatory follicle is a large antral right before ovulation and is characterised by the oocyte and cumulus cells around it being surrounded by antral space from all sides, apart from a thin stalk of cumulus cells adhering it to the follicle. These large growing follicles were enumerated independently and in combination: "all antral" being small, large antral and pre-ovulatory; and "all big follicles" including all antral and secondary follicles.

g. Unhealthy follicles were also quantified and classified according to defined morphologies and put into categories:

i. atretic (>10% pycnotic GCs or damaged oocyte), multi-nuclear, vacuoles present, enlarged oocyte with undifferentiated GCs, biovular follicles, and zona pellucida remnants (ZPRs), with representative images shown in Figure EV3.

ii. All of these, apart from ZPRs, were grouped as "large unhealthy follicles" and were counted across the whole section.

iii. Small unhealthy follicles were counted on the grid (as described above) in the same manner as healthy small follicles. They were defined by missing oocytes, or ZPRs still surrounded by a single layer of GCs. Depending on

the GCs (squamous or cuboidal) they were described as unhealthy primordial, transitory or primary.

 iv. It is noted that in some cases, due to the morphology of some unhealthy follicles there was less certainty from the blinded assessor as to whether some unhealthy follicles should be classified as primordial or transitory. When these data were re-analysed under the conditions of having all primordial and transitory follicles merged into one category, there was an identical trend for the impact of NMN co-treatment with DOX versus DOX alone (Bonferroni adjusted $p = 0.0339$), which was similar to findings in the primordial group where these stages were separated into primordial and transitory ($p = 0.0304$), as shown in Fig. 4A.

## Tumour studies

Orthotopic xenograft studies were performed in two independent laboratories with some slight modifications between studies, with results from both sets of studies presented in Fig. 7. In both studies, tumour size was measured by an investigator blinded to the treatment groups. NMN drinking water (2 g/L) was changed twice per week. Tumour volume for both studies was calculated using the formula: volume = width$^2$ * (length/2).

 Lab 1 experimental details:

(1) In Lab 1 (Fig. 7A–D), MDA-MB-231 cells from a highly aggressive, invasive and poorly differentiated triple-negative breast cancer (TNBC) cell line, were maintained in Dulbecco's modified Eagle media, Nutrient Mixture F12 (DMEM/F-12 media) with foetal bovine serum (5%), glutamine (2 mM), sodium pyruvate (1 mM) and non-essential amino acids (0.02 mM) at 37 °C in a humidified atmosphere of 5% $CO_2$.
(2) At implantation, cells were harvested by trypsin, washed in PBS, and resuspended in 20% matrigel at a density of $10 \times 10^6$ cells/ml.
(3) Five- to six-week-old athymic nude mice were ear-tagged for individual identification, and 50 μL cell suspension was injected into the fourth mammary gland of each animal, to achieve $5 \times 10^6$ cells per mouse.
(4) At the time of tumour implantation, half the animals received NMN in drinking water (2 g/L, changed twice weekly) for the remainder of the experiment.
(5) Tumour size was recorded three times per week using digital callipers.
(6) Once a cohort of 80 animals (including 40 animals receiving NMN) had tumours that reached 60–160 mm$^3$ in size, animals were randomly sorted into receiving either doxorubicin, cisplatin or vehicle control, for six groups (ctrl, doxorubicin, cisplatin, NMN alone, doxorubicin with NMN, cisplatin with NMN) with 10 animals in each group.
(7) Animals received doxorubicin (3 mg/kg), cisplatin (2 mg/kg) or a vehicle control twice per week, with animals being co-treated with NMN in drinking water also receiving a co-injection of NMN (200 mg/kg) at the same time as chemotherapy.
(8) Tumour size was measured three times per week, and animals were euthanased when tumours reached 2000 mm$^3$ in size.

 Lab 2: In this study, there was an increased rate of euthanasia for the Dox+NMN group, however, these were events unrelated to

tumour size, and were primarily related to moribund behaviour, with detailed observations are available in supplied data appendix.

(1) In Lab 2, MDA-MB-231 cells were propagated as above, and then injected into the flanks of SCID mice in the absence of a Matrigel support.
(2) Animals received doxorubicin (3 mg/kg), cisplatin (2 mg/kg) or a vehicle control four times per week, with animals being co-treated with NMN in drinking water also receiving a co-injection of NMN (200 mg/kg) at the same time as chemotherapy.
(3) Tumour size was measured three times per week, and animals were euthanised for signs of ill health, when tumours reached 2000 mm$^3$ in size, or until a pre-designated time of 24 days following tumour implantation.

## Cell survival assays

(1) Cells were seeded in 96 well plates at 10,000 cells/well in DMEM + 10% FBS and incubated overnight at 37 °C and 5% $CO_2$.
(2) NMN co-treatment was conducted by adding 90 μL of 250/500 μM NMN for four hours prior to chemotherapy treatment, and this was performed by adding 20 μL from a series of descending drug concentrations to the cell culture plate with either bleomycin, cisplatin, doxorubicin, etoposide, gemcitabine, methotrexate, paclitaxel or vincristine.
(3) Cells were returned to the incubator for 48 h, and in some cases, chemotherapy incubation times were modified to timepoints between 24 and 72 h when cells were either too resistant or too sensitive to chemotherapy to obtain a survival curve.
(4) The crystal violet staining protocol was conducted in accordance with previous methods (Feoktistova et al, 2016). Briefly, a 0.5% w/v crystal violet fixing/staining solution was produced in 50% water/methanol.
(5) Following the chemotherapy incubation time, plates were washed with PBS prior to the addition of 30 μL of the crystal violet fixing/staining solution.
(6) Plates were washed twice in tap water and dried overnight at room temperature.
(7) The stain was solubilised through the addition of 100 μL 1% SDS and homogenised on a plate shaker prior to reading absorbance at 570 nM.
(8) Cell survival curves were generated by first subtracting baseline absorbance values from the results, and normalising data to the vehicle control value as a representation of 100% viability.

## Metabolomics

(1) Animals received standard acidified drinking water with or without the addition of NMN (2 g/L) for 72 h prior to a single intraperitoneal dose of doxorubicin (10 mg/kg), with a separate i.p. injection of NMN (200 mg/kg) or saline vehicle, for a $2 \times 2$ study design.
(2) Six hours after these injections, animals were euthanised by cervical dislocation, ovaries were rapidly dissected, cleared of fat and connective tissue, and snap frozen in liquid nitrogen.

(3) To prepare samples for mass spectrometry, ovaries were sonicated in 250 μL pre-cooled (−30 °C) 80% methanol twice for 30 s.

(4) Samples were then stored at −30 °C for 20 min followed by centrifugation at $14,000 \times g$ for 10 min, 4 °C to precipitate proteins and cell debris.

(5) Supernatants were collected for LC-MSMS, with the protein pellet retained for BCA assay to determine protein concentration.

 a. These pellets were resuspended in 250 μl RIPA PMSF mix by pipetting, followed by vortexing. Samples were left to rest for 5 min, and then centrifuged for 5 min at $14,000 \times g$ to remove insoluble material. From the supernatant, 30 μl was used for the BCA assay (Thermo Fisher Scientific, Pierce™ BCA Protein Assay Kit 23225).

(6) $NAD^+$ metabolites were then measured by targeted mass spectrometry as previously described (Aflatounian et al, 2022; Bustamante et al, 2017). Data were adjusted to deuterated internal standards, then normalised to protein concentration.

## Statistics

Experiments were planned as $2 \times 2$ factorial designs, with data subject to mixed linear model analysis in $R$ (version 4.2.1) and tests for the impact of NMN treatment within chemotherapy or vehicle control calculated by estimated marginal means using base $R$ and the package *emmeans* (version 1.8.0), with *p*-values then subject to a Bonferroni correction for multiple comparisons. Prior to analysis, data were subject to outlier detection using an $R$ implementation (available on our Mendeley data site) of the ROUT method (Motulsky and Brown, 2006), using a highly conservative threshold of Q = 0.1. Across the entire investigation, this resulted in the exclusion of data from one cisplatin-treated animal from the breeding trial shown in Fig. 3, with complete datasets including from this outlier available on our Mendeley data site. Data are summarised in figures as Tukey boxplots, showing the 25–75% interquartile range with whiskers indicating 95% confidence intervals, mean values indicated by a line within boxplots. Longitudinal tumour growth was assessed using a joint modelling approach from the *Jmbayes2* package (version 0.3-1), which incorporates survival data and mixed linear modelling for longitudinal tumour growth to account for missing not at random (MNAR) data at timepoints where animals have been removed from the study due to ethical endpoints related to maximum tumour size. Detailed, annotated $R$ scripts and raw CSV files used to generate statistical analyses and generate figures have been uploaded as supplementary files, allowing reproducible analysis—please see Data availability section.

## Study design

This study was conducted in accordance with the ARRIVE guidelines on the use of animals in research, with an attached checklist.

The experimental unit for all experiments were data from individual animals, including oocyte yield per animal (Fig. 1), reproductive output per female (Figs. 2 and 3), ovarian reserve per ovary (Fig. 4), metabolomics (Fig. 5), proteomics (Fig. 6) and tumour growth (Fig. 7A–H).

The inclusion criteria for the study were animals ordered from our external supplier at the ages indicated in methods as described above. Exclusion criteria were if animals had to be removed from the study for ethical reasons due to poor health unrelated to the intervention. As described above, ROUT outlier analysis was applied to all data using a cut-off of Q = 0.1. Across the entire study, data from two animals were excluded: one outlier from the cisplatin alone group in the breeding trial shown in Fig. 3, and one animal that had to be removed from the cisplatin+NMN group also in the breeding trial shown in Fig. 3. Both exclusions are described in the figure legend (Fig. 3).

Animals were randomised into chemotherapy or vehicle treatment based on the order that they were first picked up out of the cage, alternating between the two for the next animal. Cages were similarly randomised into NMN or vehicle treatment in their drinking water based on the order they were taken from the rack, alternating between each intervention. This strategy of alternating treatments between each subsequent animal or cage was used to minimise the likelihood of confounding effects.

As described in previous sections, investigators were blinded to treatment groups at the time of oocyte collection, assessing breeding output, counting follicles, and monitoring tumour growth. Cages were labelled with animal numbers, and treatment groups correlating to animal numbers were recorded in a separate lab book. To maintain blinding, a separate investigator administered doxorubicin injections, due to the obvious bright red colour of this compound.

Data were analysed by mixed linear model analysis followed by post hoc analysis of estimated marginal means, as described in the statistics section above. We did not adjust this model for data that did not meet tests for normality, as mixed linear models are generally robust to violations of this assumption (Knief and Forstmeier, 2021).

Sample size was determined based on precedents from previous papers in the field that have used similar primary outcomes in models of chemotherapy induced infertility (Ben-Aharon et al, 2010; Goldman et al, 2017; Kalich-Philosoph et al, 2013; Kerr et al, 2012).

The primary outcome of each experiment is described in the relevant figure legend, with additional detail in the methods section above.

## Proteomics

### Sample preparation

(1) Proteomics was performed using the contralateral ovary from the same animals used for metabolomics as described in the previous section, where samples were collected six hours following a single dose of doxorubicin (10 mg/kg) or vehicle control and a separate i.p. injection with NMN (200 mg/kg) or saline vehicle.

(2) Rapidly dissected ovaries were snap frozen in liquid nitrogen and stored at −80 °C prior to processing.

(3) Pre-cooled (−30 °C) methanol (250 μL, 80%) was added to each ovary, which was then sonicated twice for 30 s while sitting on ice.

(4) Samples were incubated at $-30\,°C$ for 20 min, following which they were centrifuged at $14{,}000 \times g$ for 10 min at $4\,°C$.

(5) Protein pellets were resuspended and vortexed in 250 μL fresh RIPA buffer (150 mM NaCl, 1% Igepal CA-630, 0.5% sodium deoxycholate, 0.1% SDS, 50 mM Tris, pH 8.0) with 0.1 mM PMSF, following which they were incubated at room temp for 5 min.

(6) Samples were then centrifuged for 5 min at $14{,}000 \times g$, and aliquot taken for protein assays using the BCA assay as above, which was used to normalise protein extracts to a uniform concentration.

(7) Samples were then dried using a SpeedVac Plus (Savant, SC110A) and resuspended in 10 μL deionized water with 2% (v/v) acetonitrile, 0.05% (v/v) heptafluorobutyric acid (HFBA) and 0.01% (v/v) formic acid.

(8) Samples were reduced (5 mM DTT, 37 °C, 5 min) and alkylated (10 mM iodoacetamide at room temp, 30 min). Samples were then subject to trypsin digestion and clean-up using the SP3 method (Hughes et al, 2021).

### Mass spectrometry

Digest peptides were separated by nanoLC using an Ultimate nanoRSLC UPLC and autosampler system (Dionex, Amsterdam, Netherlands). Samples (2.5 μL) were concentrated and desalted onto a micro C18 precolumn (300 μL × 5 mm, Dionex) with $H_2O{:}CH_3CN$ (98:2, 0.2% TFA) at 15 μL/min. After a 4 min wash the pre-column was switched (Valco 10 port UPLC valve, Valco, Houston, TX) into line with a fritless nano column (75 μm x ~ 15 cm) containing C18AQ media (1.9 μL, 120 Å Dr Maisch, Ammerbuch-Entringen Germany). Peptides were eluted using a linear gradient of $H_2O{:}CH_3CN$ (98:2, 0.1% formic acid) to $H_2O{:}CH_3CN$ (64:36, 0.1% formic acid) at 200 μL/min over 30 min. High voltage (2000 V) was applied to low volume Titanium union (Valco) and the tip positioned ~5 mm from the heated capillary ($T = 275\,°C$) of an Orbitrap Fusion Lumos (Thermo Electron, Bremen, Germany) mass spectrometer. Positive ions were generated by electrospray and the Fusion lumos operated in data-dependent acquisition mode (DDA).

Proteomics were performed as previously described (Michalski et al, 2011). Briefly, a survey scan $m/z$ 350–1750 was acquired in the orbitrap (resolution = 120,000 at $m/z$ 200, with an accumulation target value of 400,000 ions) and lockmass enabled ($m/z$ 445.12003). Data-dependent tandem mass spectrometry (MS) analysis was performed using a top-speed approach (cycle time of 2 s). MS2 spectra were fragmented by higher energy collision dissociation (HCD) (normalised collision energy (NCE) = 30) activation mode, and the ion-trap was selected as the mass analyser. The intensity threshold for fragmentation was set to 25,000. A dynamic exclusion of 20 s was applied with a mass tolerance of 10 ppm.

Peptide ions ($>2.5 \times 10^4$ counts, charge states +2 to +5) were sequentially selected for MS/MS (also known as tandem MS) using data dependent acquisition, with the total number of dependent scans maximized within 2 s cycle times. Product ions were generated via higher energy collision dissociation (collision energy = 30; maximum injection time = 250 ms; multi-stage MS (MSn) automatic gain control (AGC) = $5 \times 10^4$; inject ions for all available parallelizable time enabled) and mass analysed in the linear ion trap. Dynamic exclusion was enabled and set to:

$n$ times = 1, exclusion duration 20 s, ±10 ppm. The default setting was used in the MaxQuant set up, with label-free quantitation and match between-run features turned on.

### Data analysis

(1) Mass spectrometry data was first analysed using the MaxQuant package v1.6.17 (Cox and Mann, 2008), which was integrated with the Andromeda peptide database for protein identification.

(2) The "proteinGroups.txt" file (available on our open data site) was exported from MaxQuant, imported into R and analysed using a local copy of the LFQ-Analyst package v1.2.3 (Shah et al, 2020).

(3) Data were filtered to remove proteins that were identified by only a single peptide, as well as those which were not present in all samples of at least one treatment group.
   a. Missing values were imputed using the k Nearest Neighbours algorithm, which was selected due to previous work suggesting this is the most accurate strategy for imputation (Lazar et al, 2016).
   b. Cut-offs for significance were set to an adjusted $p$-value of 0.05, and a log2 fold change of 1, with significance subject to Benjamini–Hochberg FDR correction.
   c. This package tests for differences between groups using empirical Bayes statistics and linear models provided by the *limma* package.

## Cardiac function

To test the effects of NMN on DOX-induced cardiotoxicity, we used C57BL6 male mice treated with DOX weekly for four weeks, as described previously (Dolinsky et al, 2013).

(1) DOX was prepared at 2.16 mg/ml in a vehicle of 4.32% DMSO in saline, with 100 μL per 27 g body weight, achieving a dose of 8 mg/kg, with control animals instead receiving a DMSO saline vehicle. NMN-treated groups had NMN added to the above solutions at 13.5 mg/100 μL to achieve a final dosing of 500 mg/kg, in addition to NMN in drinking water (2 g/L) which started 1 day prior to the first dose of DOX. The overall cumulative dose of DOX at the end of the 4-week period was 32 mg/kg, which is the human equivalent dose of 96 mg/$m^2$.

(2) Due to the bright red colour of doxorubicin, weekly injections were performed by investigators who were not involved in subsequent imaging or image analysis, in order to preserve blinding to treatment groups.

(3) Echocardiography was performed on a weekly basis under blinded conditions using the Vevo® 2100 ultrasound system (Visualsonics, Toronto, ON, Canada) as previously described (Guenancia et al, 2015).
   a. One day prior to imaging, all mice were lightly anaesthetized with isoflurane (3% for induction and 1–1.5% for maintenance) in 100% oxygen, and the chest was shaved using a chemical hair remover (Veet, Reckitt Benckiser, NSW, AU).
   b. The following day under the same anaesthetic conditions, mice were placed onto a heated platform in a supine position with paws taped down onto pads for heart rate monitoring.

c. The platform and a 30 MHz transducer probe was adjusted at approximately 30° and 10° counter-clockwise, to obtain the correct orientation of the left ventricle (LV) in parasternal long axis (PSLA) and parasternal short axis (PSA) views at the level of the papillary muscles.

d. From each of these views, 2-dimensional images in separate modes, B-mode and M-mode, were recorded for analysis and quantification.

(4) The Vevo® LAB software (Version 1.7.1) was used for quantification of the recorded images and involved physical tracings of the LV endocardium in PSA M-mode to obtain cardiac parameters that measure systolic function.

a. The main measure of cardiac function (or dysfunction) includes ejection fraction (EF) and fractional shortening (FS) as well as diastolic and systolic LV muscle diameters of the interventricular septal wall (IVS), internal diameter (LVID) and posterior wall (LVPW). Each calculation was measured as an average of four replicates ± SD.

b. Data are presented here (Fig. EV5) as a percentage change of each animal from its baseline readings.

## Graphics

Data figures were generated in R using the packages *ggplot2* (3.4.0), *ggtext* (0.1.2), *ggpubr* (0.4.0), *ggsignif* (0.6.3) and *ggh4x* (0.2.3.9000). Data are shown as Tukey boxplots, with all raw datapoints overlaid. The illustrations in Figs. 1A,G, 2A, 3A, and 5I,J were generated under license using BioRender.com. Chemical structures in Fig. 5J were prepared using ChemDraw (v19.0, PerkinElmer).

## For more information

- Future Fertility network - http://www.futurefertility.com.au
- Oncofertility Consortium - https://oncofertility.msu.edu
- Alliance for Fertility Preservation - https://www.allianceforfertilitypreservation.org
- Fertility & Research Centre (FRC) https://www.fertilityandresearchcentre.com.au
- ESHRE Fertility Preservation Group https://www.eshre.eu/Specialty-groups/Special-Interest-Groups/Fertility-Preservation
- Youth Cancer Europe - https://www.youthcancereurope.org/cancer-fertility-a-youth-cancer-europe-advocacy-project/
- Dr Lindsay Wu – https://www.unsw.edu.au/staff/lindsay-wu
- Prof Robert Gilchrist – https://www.unsw.edu.au/staff/robert-gilchrist
- Prof Hayden Homer https://clinical-research.centre.uq.edu.au/profile/1350/hayden-homer.

## Data availability

All raw data have been uploaded to our Mendeley data site (https://doi.org/10.17632/fs5y6ggskv.1) as .csv and .txt files, with annotated R scripts for processing data, statistical analysis and generating figures. For histological assessment of ovarian reserve, all sections were digitally scanned for follicle quantitation, however, each of these scans resulted in large files which have not been uploaded due

### The paper explained

#### Problem

While the overall rates of survival for cancer patients have improved, many patients experience long-term health issues due to the side effects of chemotherapy drugs that are used in their treatment. In women, this includes the risk of infertility and premature menopause, due to the toxicity of chemotherapy towards the ovaries. To manage this, chemotherapy treatment in cancer patients can be delayed allowing oocytes to be collected and frozen, in order to provide future options for fertility. This is not ideal, as it requires a delay in chemotherapy treatment, and the involvement of fertility specialists who may not be available in all clinical settings. This problem is especially acute in the paediatric setting in girls with cancer, as the hormonal stimulation needed for oocyte collection can induce premature menopause. In these patients, the current option is to have a surgeon collect an ovarian biopsy to be frozen down for long-term storage, as an option to be transferred back to patients later in life. While this has allowed some childhood cancer survivors to start a family, it runs the risk of cancer recurrence due to the re-introduction of cancer cells via the biopsy, especially in the case of blood cancers. Overall, what is needed is a non-invasive, drug-based approach to protecting the ovaries against the toxic effects of chemotherapy in cancer patients.

#### Results

We previously found that declining levels of the metabolite nicotinamide adenine dinucleotide (NAD$^+$) were a cause of age-related female infertility, and that restoring these levels could improve oocyte quality and fertility in older animals. Cancer patients who previously received chemotherapy treatment experience a high rate of chronic diseases, including infertility, which together resemble a form of accelerated biological ageing. Given this, we reasoned that a similar strategy could help to protect against chemotherapy-induced infertility. We found that supplementing animals with the NAD$^+$ precursor nicotinamide mononucleotide (NMN), or genetic overexpression of the NAD$^+$ biosynthetic enzymes NMNAT1 and NMNAT3, could protect against infertility caused by the drugs doxorubicin and cisplatin. One concern is that a strategy which protects against chemotherapy-induced damage in the ovaries could also impact the efficacy of chemotherapy against tumours, however, we found that NMN treatment did not do this in a xenograft model of breast cancer, and in fact slowed tumour growth.

#### Impact

This work suggests that supplementation with naturally occurring NAD+ precursors could improve long-term quality of life in women with cancer, through protecting ovaries and overall fertility against the effects of commonly used, toxic, but necessary, chemotherapy drugs. In our studies in mice, this approach did not reduce the effectiveness of chemotherapy against tumours, suggesting that this approach could be safe to use at the same time as chemotherapy drugs. While promising, it should be emphasised that our studies were in mice, and further work will be needed to test the safety and efficacy of this approach in patients.

to file size limits on open data servers. These have been maintained for long-term storage on UNSW servers and are available upon request.

The source data of this paper are collected in the following database record: biostudies:S-SCDT-10_1038-S44321-024-00119-w.

## Peer review information

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

## Acknowledgements

The salary of LEW is supported from a Hevolution/American Federation for Aging Research (AFAR) New Investigator award. This work was supported by the National Health and Medical Research Council (NHMRC) of Australia, through grants APP1103689 and APP1122484 to LEW, DAS and HAH, APP1139763 to RBG, LEW and KAW, a Career Development Fellowship to LEW (APP1127821) and Senior Research and Investigator Fellowships to RBG (APP1117538 and APP2009940). The salary and experimental costs of MB and DMG working in the lab of RBG and LEW were partly supported by Jumpstart Fertility. KS was an employee of Jumpstart Fertility. Orthotopic xenograft tumour models were performed at external contract research organisations under funding from Jumpstart Fertility and Metro Biotech. We gratefully acknowledge assistance from the UNSW Biological Resource Centre and Tzong-Tyng Hung from the Biological Resource Imaging Laboratory (BRIL) of the Mark Wainwright Analytical Centre, UNSW, which is a member of the National Imaging Facility, and the Bioanalytical Mass Spectrometry Facility of the Mark Wainwright Analytical Centre at UNSW. We also wish to thank the Solina Chau Foundation and Mr Hejun (Steven) Zhang for their philanthropic support.

## Author contributions

**Wing-Hong Jonathan Ho**: Conceptualization; Investigation. **Maria B Marinova**: Validation; Investigation. **Dave R Listijono**: Investigation. **Michael J Bertoldo**: Supervision; Investigation; Methodology. **Dulama Richani**: Data curation; Investigation; Methodology. **Lynn-Jee Kim**: Investigation. **Amelia Brown**: Investigation. **Angelique H Riepsamen**: Investigation. **Safaa Cabot**: Investigation. **Emily R Frost**: Writing—review and editing. **Sonia Bustamante**: Investigation; Methodology. **Ling Zhong**: Investigation; Methodology. **Kaisa Selesniemi**: Investigation. **Derek Wong**: Investigation. **Romanthi Madawala**: Investigation. **Maria Marchante**: Investigation. **Dale M Goss**: Investigation. **Catherine Li**: Investigation. **Toshiyuki Araki**: Resources. **David J Livingston**: Validation. **Nigel Turner**: Resources. **David A Sinclair**: Supervision; Writing—review and editing. **Kirsty A Walters**: Supervision; Validation; Methodology. **Hayden A Homer**: Conceptualization; Supervision; Funding acquisition; Methodology; Project administration; Writing—review and editing. **Robert B Gilchrist**: Conceptualization; Supervision; Funding acquisition; Methodology; Project administration; Writing—review and editing. **Lindsay E Wu**: Conceptualization; Data curation; Formal analysis; Supervision; Funding acquisition; Visualization; Writing—original draft; Project administration; Writing—review and editing.

Source data underlying figure panels in this paper may have individual authorship assigned. Where available, figure panel/source data authorship is listed in the following database record: biostudies:S-SCDT-10_1038-S44321-024-00119-w.

## Disclosure and competing interests statement

RBG is a consultant to City Fertility CHA Global and is a Scientific Advisory Board member to Cooper Surgical. MJB is currently an employee of Ferring Pharmaceuticals, which produce drugs used in reproductive medicine. LEW, HAH and DAS are co-founders, shareholders, directors and advisors of Jumpstart Fertility Inc, which was founded to develop NAD$^+$ precursors for the treatment of age-associated female infertility. The salaries of MJB and DG were paid in part by contract research from Jumpstart Fertility to UNSW. KS was an employee of Jumpstart Fertility. DJL is an employee and shareholder in Metro International Biotech, which is developing NAD$^+$ precursors as therapeutics for a range of conditions. LEW and DAS are advisors and shareholders in EdenRoc Sciences, the parent company of Metro Biotech NSW and Metro Biotech, and in Life Biosciences LLC and its daughter companies. DAS is also a founder, equity owner, advisor, director, consultant, investor and/or inventor on patents licensed Cohbar, Galilei Biosciences, Tally Health,

EdenRoc Sciences, MetroBiotech, and Life Biosciences. He is an inventor on a patent application filed by Mayo Clinic and Harvard Medical School that has been licensed to Elysium Health. For a full list and details see https://genetics.med.harvard.edu/sinclair/. MMC is currently an employee of Gameto Inc., which is developing cell-based therapies for reproductive medicine.

# Expanded View Figures

**Figure EV1.   Kinetics of oocyte maturation.**

Germinal vesicle (GV) stage cumulus oocyte complexes (COCs) from animals stimulated with PMSG in main Fig. 1 were collected in media containing IBMX to prevent meiotic maturation, mechanically denuded and moved to IBMX-free media to allow meiotic resumption. The proportion of oocytes undergoing germinal vesicle breakdown (GVBD) and polar body extrusion (PBE) were assessed at the indicated timepoints in oocytes obtained from mice treated with (**A**) doxorubicin (Dox) or (**B**) cisplatin in the presence or absence of NMN, as in main Fig. 1B,C, and in (**C**) NMNAT1 and (**D**) NMNAT3 transgenic or wild-type (WT) littermates treated with Dox, as in main Fig. 1D,F.

▶

                                                                           

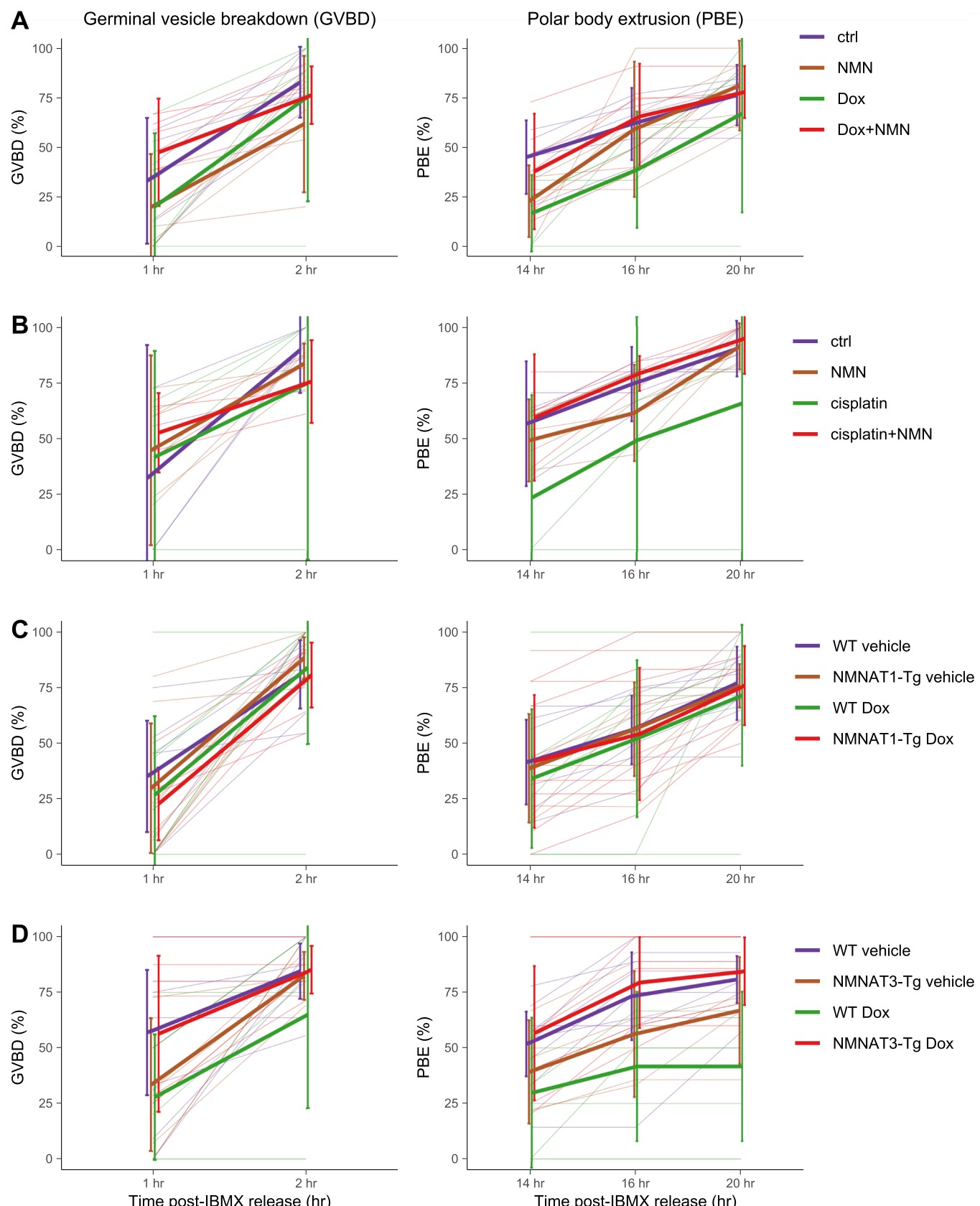

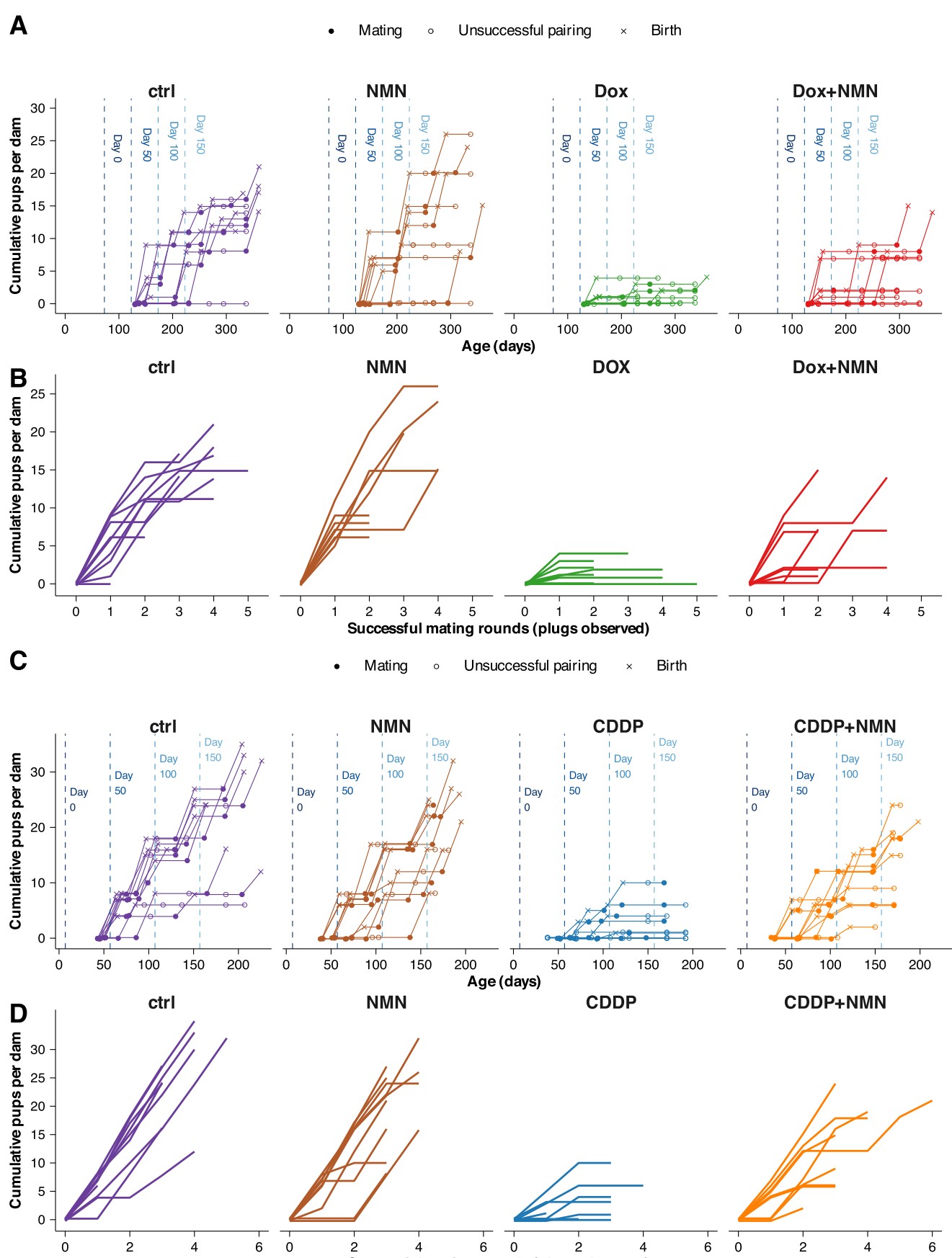

Figure EV2.   Individualised breeding trial data from Figs. 2 and 3.

(A, B) Breeding trial data from doxorubicin (Dox) treatment as per Fig. 2A, or (C, D) from cisplatin (CDDP) treatment as per Fig. 3A. Each line represents the cumulative number of pups from each dam, displayed (A, C) as a function of time, showing each successful or unsuccessful mating (confirmed by observation of a vaginal plug) and births, as summarised for the entire cohorts in Figs. 2B and 3B. The timing of dox or cisplatin treatment is indicated as Day 0 by a dark blue line, with lighter dashed lines for subsequent 50-day increments. Results are also expressed as (B, D) pups per dam per round of confirmed mating, as confirmed by the presence of a vaginal plug.

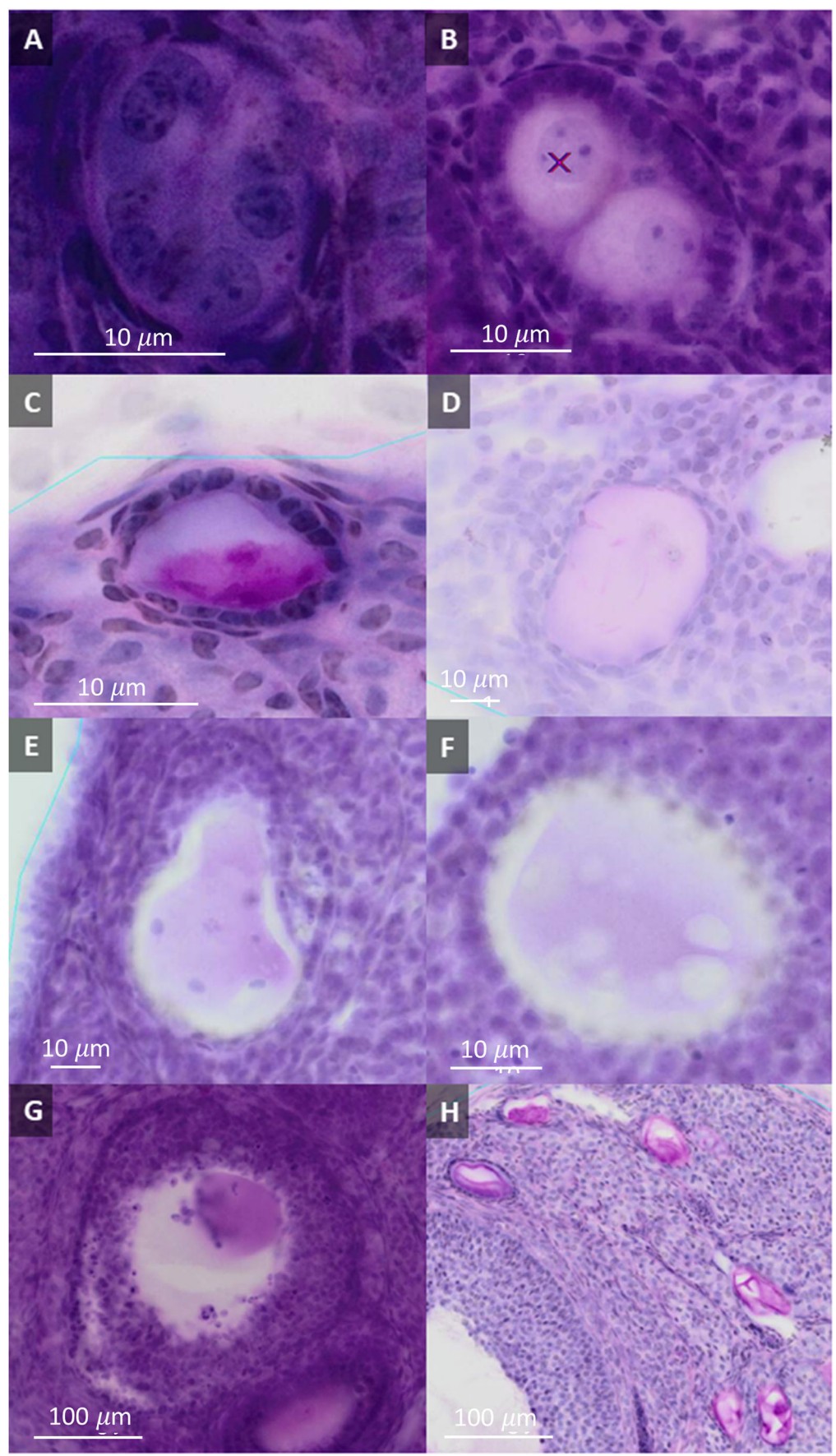

◀ **Figure EV3.  Representative images of follicles classified as unhealthy during assessments of ovarian reserve (Fig. 3).**

Images show follicles with (**A**) no oocyte; (**B**) biovular follicle; (**C**) small follicle with zona pellucida remnants (ZPRs); (**D**) enlarged oocyte with undifferentiated GCs; (**E**) multinuclear oocyte; (**F**) vacuoles present in oocyte; (**G**) atretic follicle; (**H**) ZPRs. Scale bars are as indicated in each image.

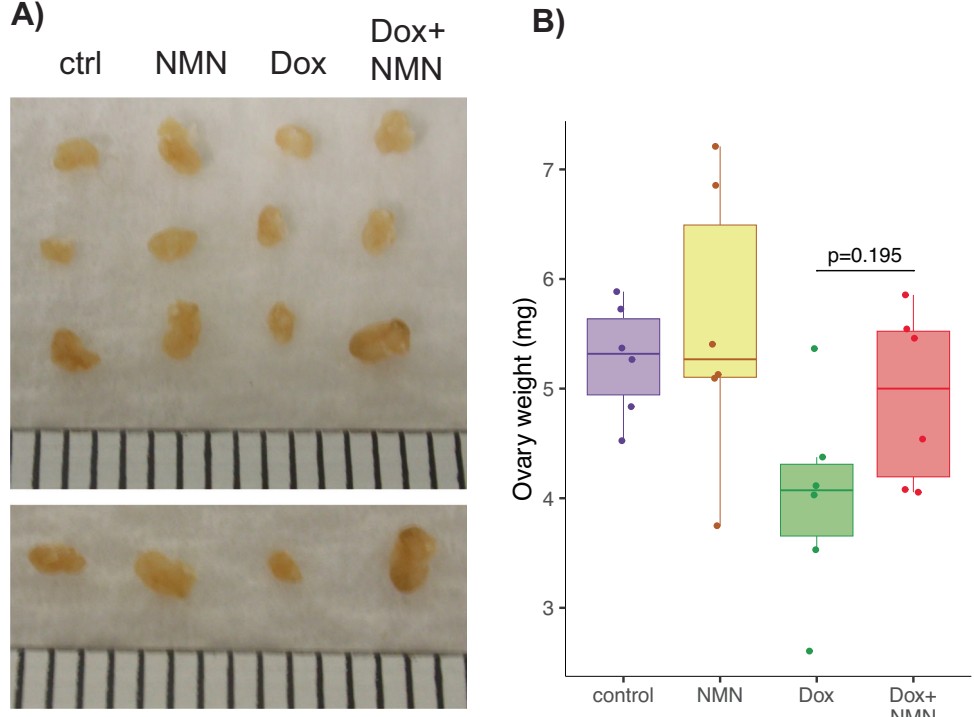

**Figure EV4. Whole ovary weight and appearance.**

(A) Macroscopic appearance of ovaries from non-PMSG stimulated animals treated with doxorubicin (Dox) in the presence or absence of NMN, collected 8 weeks after chemotherapy as described in Figs. 1, 2 and 4. Rulers in pictures show length in millimetres. (B) Weights of ovaries collected for stereology from animals treated with doxorubicin (Dox) in the presence or absence of NMN, as described in Fig. 4. $N = 5$–6 animals per treatment. Data were analysed by estimated marginal means for NMN from a linear model of DOX and NMN treatment with Bonferroni correction, p-values as indicated.

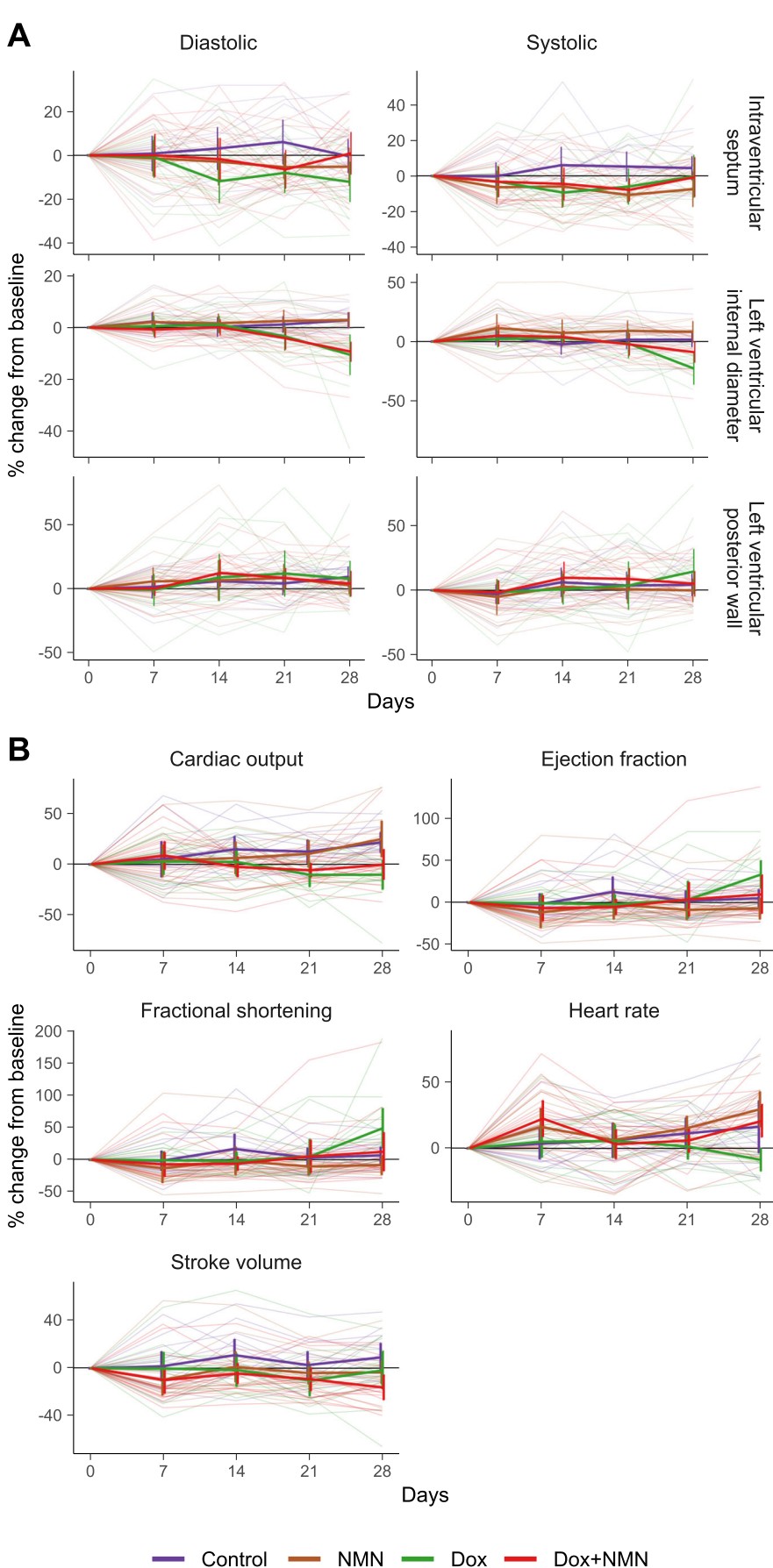

◀ **Figure EV5. Doxorubicin (DOX) induced cardiotoxicity.**

Animals were treated with DOX (8 mg/kg) once per week for 4 weeks, with ultrasound imaging every week to assess parameters of (**A**) left ventricular structure and (**B**) cardiac function. Thick lines indicate the mean values for each treatment group, thin transparent lines indicate data from individual animals. $n = 13$–15 per group, error bars are mean ± 95% CI.

