## [Peer Review File · EMBO Molecular Medicine]

Fertility protection during chemotherapy treatment by boosting the NAD(P)⁺ metabolome

Wing-Hong Jonathan Ho, Maria Marinova, Dave Listijono, Michael Bertoldo, Dulama Richani, Lynn-Jee Kim, Amelia Brown, Angelique Riepsamen, Safaa Cabot, Emily Frost, Sonia Bustamante, Ling Zhong, Kaisa Selesniemi, Derek Wong, Romanthi Madawala, Maria Marchante, Dale Goss, Catherine Li, Toshiyuki Araki, David Livingston, Nigel Turner, David Sinclair, Kirsty Walters, Hayden Homer, Robert Gilchrist, and Lindsay Wu

Corresponding author: Lindsay Wu (lindsay.wu@unsw.edu.au)

Review Timeline:

Submission Date:	17th Apr 24
Editorial Decision:	8th May 24
Revision Received:	25th Jun 24
Editorial Decision:	12th Jul 24
Revision Received:	24th Jul 24
Accepted:	30th Jul 24

Editor: Lise Roth

Transaction Report:

8th May 2024

Dear Dr. Wu,

Thank you for the submission of your manuscript to EMBO Molecular Medicine. We have now received feedback from the three reviewers who agreed to evaluate your manuscript. As you will see from the reports below, the referees acknowledge the interest of the study and are overall supporting publication of your work pending appropriate revisions.

We further consulted with the referees, and after discussion with my colleagues, we agreed that single cell transcriptomic experiments (referee #1) will NOT be required for further consideration in our journal.

We would encourage you to further elucidate experimentally the underlying mechanism, however this could be complemented by adequate discussion making clear what the outstanding questions are in the absence of experimental answers.

Addressing the other reviewers' concerns will be necessary for further considering the manuscript in our journal, and acceptance of the manuscript will entail a second round of review.

EMBO Molecular Medicine encourages a single round of revision only and therefore, acceptance or rejection of the manuscript will depend on the completeness of your responses included in the next, final version of the manuscript. For this reason, and to save you from any frustrations in the end, I would strongly advise against returning an incomplete revision.

We are expecting your revised manuscript within three months, if you anticipate any delay, please contact us.

We require:

4) A .docx formatted letter INCLUDING the reviewers' reports and your detailed point-by-point responses to their comments. As part of the EMBO Press transparent editorial process, the point-by-point response is part of the Review Process File (RPF), which will be published alongside your paper.

5) A complete author checklist, which you can download from our author guidelines (<https://www.embopress.org/page/journal/17574684/authorguide#submissionofrevisions>). Please insert information in the checklist that is also reflected in the manuscript. The completed author checklist will also be part of the RPF.

6) It is mandatory to include a 'Data Availability' section after the Materials and Methods. Before submitting your revision, primary datasets produced in this study need to be deposited in an appropriate public database, and the accession numbers and database listed under 'Data Availability'. Please remember to provide a reviewer password if the datasets are not yet public (see <https://www.embopress.org/page/journal/17574684/authorguide#dataavailability>).

7) For data quantification: please specify the name of the statistical test used to generate error bars and P values, the number (n) of independent experiments (specify technical or biological replicates) underlying each data point and the test used to calculate p-values in each figure legend. The figure legends should contain a basic description of n, P and the test applied. Graphs must include a description of the bars and the error bars (s.d., s.e.m.). Please provide exact p values.

8) Our journal encourages inclusion of *data citations in the reference list* to directly cite datasets that were re-used and obtained from public databases. Data citations in the article text are distinct from normal bibliographical citations and should

directly link to the database records from which the data can be accessed. In the main text, data citations are formatted as follows: "Data ref: Smith et al, 2001" or "Data ref: NCBI Sequence Read Archive PRJNA342805, 2017". In the Reference list, data citations must be labeled with "[DATASET]". A data reference must provide the database name, accession number/identifiers and a resolvable link to the landing page from which the data can be accessed at the end of the reference. Further instructions are available at .

9) We replaced Supplementary Information with Expanded View (EV) Figures and Tables that are collapsible/expandable online. A maximum of 5 EV Figures can be typeset. EV Figures should be cited as 'Figure EV1, Figure EV2" etc... in the text and their respective legends should be included in the main text after the legends of regular figures.

10) The paper explained: EMBO Molecular Medicine articles are accompanied by a summary of the articles to emphasize the major findings in the paper and their medical implications for the non-specialist reader. Please provide a draft summary of your article highlighting

11) For more information: There is space at the end of each article to list relevant web links for further consultation by our readers. Could you identify some relevant ones and provide such information as well? Some examples are patient associations, relevant databases, OMIM/proteins/genes links, author's websites, etc...

12) Author contributions: CRediT has replaced the traditional author contributions section because it offers a systematic machine readable author contributions format that allows for more effective research assessment. Please remove the Authors Contributions from the manuscript and use the free text boxes beneath each contributing author's name in our system to add specific details on the author's contribution. More information is available in our guide to authors.

13) Disclosure statement and competing interests: We updated our journal's competing interests policy in January 2022 and request authors to consider both actual and perceived competing interests. Please review the policy <https://www.embopress.org/competing-interests> and update your competing interests if necessary.

14) Every published paper now includes a 'Synopsis' to further enhance discoverability. Synopses are displayed on the journal webpage and are freely accessible to all readers. They include a short stand first (maximum of 300 characters, including space) as well as 2-5 one-sentences bullet points that summarizes the paper. Please write the bullet points to summarize the key NEW findings. They should be designed to be complementary to the abstract - i.e. not repeat the same text. We encourage inclusion of key acronyms and quantitative information (maximum of 30 words / bullet point). Please use the passive voice. Please attach these in a separate file or send them by email, we will incorporate them accordingly.

15) As part of the EMBO Publications transparent editorial process initiative (see our Editorial at <http://embomolmed.embopress.org/content/2/9/329>), EMBO Molecular Medicine will publish online a Review Process File (RPF) to accompany accepted manuscripts.

In the event of acceptance, this file will be published in conjunction with your paper and will include the anonymous referee reports, your point-by-point response and all pertinent correspondence relating to the manuscript. Let us know whether you agree with the publication of the RPF and as here, if you want to remove or not any figures from it prior to publication. Please note that the Authors checklist will be published at the end of the RPF.

I look forward to receiving your revised manuscript.

Yours sincerely,

Lise Roth

**** Reviewer's comments ****

Referee #1 (Remarks for Author):

Overall, the study presents an innovative approach to addressing the significant concern of chemotherapy-induced ovarian failure and infertility in female cancer patients. The exploration of pharmacological and transgenic strategies to boost the NAD⁺ metabolome and its potential protective effect on ovarian function is noteworthy. The findings regarding the amelioration of chemotherapy-induced female infertility in mice, as well as the observed reduction in tumor growth without compromising the efficacy of chemotherapy drugs in a triple-negative breast cancer mouse model, are promising. However, there are several areas that require further clarification and consideration:

1. While the study demonstrates the beneficial effects of boosting the NAD⁺ metabolome on ovarian function and cancer treatment outcomes, further elucidation of the underlying molecular mechanisms is warranted. Providing insights into how NAD⁺ precursors exert their protective effects on ovarian function and tumor growth would enhance the significance of the findings.
2. Previous studies have demonstrated a significant impact of chemotherapy drug treatment on ovarian reserve, resulting in extensive apoptosis of primordial follicles. Therefore, the dosage of chemotherapy drugs plays a crucial role. Hence, the authors need to provide justification for why this particular dosage was chosen.
3. Although the authors attempted to identify potential mechanisms through proteomic analysis, the link between differentially expressed proteins and other factors is not fully established. For example, we know that chemotherapy can induce DNA damage and apoptosis. However, whether NMN supplementation truly affects this process requires further experimental validation.
4. To clarify the mechanism of action of NMN, we suggest conducting additional experiments such as single-cell transcriptomic sequencing. This will help identify which cells and pathways are altered by NMN supplementation. Such experiments could provide valuable insights for interpreting the results more effectively.

Referee #2 (Comments on Novelty/Model System for Author):

This is a very nice study, with several lines of investigation carried out at each step, leading to a high level of robustness to the conclusions drawn. The work has the potential to have high medical impact down the line, although at this stage, work is all on the mouse model, or on cell lines.

Referee #2 (Remarks for Author):

This manuscript details the effect of a potential chemo-protectant on female fertility, that of the NADPH system. Overall, it is a very nice, highly robust study. Of particular note, the authors investigate treatment in several ways, and look at two drugs, with effects on ovarian function and fertility also examined in a range of ways, all adding to the strength of the work. It is also clearly written and comprehensively discussed.

None of my comments are major, but rather aimed at increasing the clarity for readers.

1. Authors should provide a clearer explanation of the statistical analysis carried out, and in particular show results more fully, not just comparing chemotherapy drug-exposed to the chemo + treated groups. Experiments were designed for 2x2 factorial analysis. Given that, the authors could also show if there was an effect of overall chemotherapy exposure, and of overall NADPH co-treatment, or else they could carry out more post hoc comparisons. As JUST ONE EXAMPLE of this, on p5, the authors state that there is a 'profound' loss of COC induced by DOX. Is that loss significant? What is the no p value?
2. More work was carried out examining the effect of doxorubicin than of cisplatin. (Fig 1: NMNAT1 only tested against DOX; and Figs 4-6 work). To be clear, I am not suggesting that yet further work is needed, but just that the authors discuss this,

explaining why that was so.

3. Figures could be clearer, particularly figures 1-3.

- a. Fig 1. The layout could be much clearer, for instance by grouping like panels together. Also, it would be helpful if panels were given headings - e.g. at present it is difficult to differentiate between two the different transgenic mouse lines, one small numeral on the x axis.
- b. Figs 2 and 3: it might be worth considering putting the data in panels B and D into a supplemental figure, to allow the key data to be focussed on more.
- c. Fig 4: again the grouping of panels could be clearer. D shows total follicle numbers, but presumably includes data from E and F.

4. The format of references in the text is often odd re listing of authors

Referee #3 (Remarks for Author):

In the paper "Fertility protection during chemotherapy treatment by boosting the NAD(P)⁺ metabolome", Wing-Hong Jonathan Ho and co-authors investigated the contribution of dysregulated NAD⁺ metabolism in chemotherapy-induced female infertility. The authors demonstrated that restoring NAD⁺ metabolome after chemotherapy treatments improved female fertility without impacting anti-cancer efficacy. The topic presented in the manuscript is interesting and expand our understanding of NAD⁺ metabolism in chemotherapy-induced infertility with potential translational value. Overall, the data support the conclusions stated by the authors and manuscript is generally clearly written. However, I suggest the following points to be addressed prior to publication:

MAJOR:

- While the authors show that pharmacological (NMN) or genetic (NMNAT1 or 3 tg) interventions to replenish NAD⁺ are efficacious in improving female mouse fertility, they did not investigate the role of NAD⁺ consuming enzymes in driving this phenotype. The authors mentioned a couple of times the potential role of PARPs but they did not mention other enzymes such as CD38 which has been reported as overexpressed and key in other Doxorubicin in vivo models (<https://academic.oup.com/circovasres/article/120/3/286/7589532>). Moreover, CD38 has recently gained attention in the context of ovarian aging (<https://www.sciencedirect.com/science/article/pii/S2589004223020266> , <https://www.nature.com/articles/s43587-023-00532-9>). If other NAD⁺ consuming enzymes are involved in this phenotype, specific pharmacological inhibition of these targets should be considered. The authors should comment and expand their discussion including relevant literature.
- Figure 5: the data points show a lot of variability which is expected in these types of analysis. The authors should consider increasing the n to at least n=10.
- Figure 5: the authors should comment NMN-induced metabolomic changes such as increased N-Methyl-NAM and decreased NADH.
- While Figure 5 and 6 are extremely interesting, the metabolomic and proteomic changes observed might not be relevant for the studies showed in Figure 1-4 where mice had longer NMN administration after chemotherapy treatment.
- Page 10, Supp. Fig.4: This evidence was published last year in the following paper: <https://www.ncbi.nlm.nih.gov/pmc/articles/PMC9818647/> The authors should mention and comment the published data in the context of their study.

MINOR:

- Figure 1: it is somehow difficult to follow as the sub-figures are not presented in order (e.g. Fig. 1E mentioned before Fig 1C).
- Figure 7 D and H are low quality and hard to read
- I would use the same name in stats legends and axis (e.g. Fig 5: use Dox instead of chemo).

Ho and Marinova et al, EMM-2024-19836 – Response to reviewers

We sincerely thank all the referees for their efforts, and for providing comments that have considerably improved our manuscript. We have addressed each of these comments below, with new text that has been added to the manuscript shown in italics. While we have not been able to address every single one of these suggestions, we hope that our responses will overall address the reviewer's comments. In some cases, we will defer to the Editor for advice, particularly around whether some panels should be moved from the main figures (Figs. 2, 3) to Extended View.

Please note that we are now taking advantage of EMBO Molecular Medicine's Extended View format for figures that would otherwise go to supplementary data, resulting in a slight re-arrangement in the order of those figures. These are indicated in the text as Figure EV1, Figure EV2 etc.

Finally, we would welcome input from the reviewers on one additional point: EMBO Molecular Medicine offers a "for more information" section at the end of a manuscript that allows links to external resources relevant to the paper, including for example patient advocacy organisations. If the Reviewers are aware of any additional organisations that we should list here, please let us know.

Referee #1 (Remarks for Author):

Overall, the study presents an innovative approach to addressing the significant concern of chemotherapy-induced ovarian failure and infertility in female cancer patients. The exploration of pharmacological and transgenic strategies to boost the NAD⁺ metabolome and its potential protective effect on ovarian function is noteworthy. The findings regarding the amelioration of chemotherapy-induced female infertility in mice, as well as the observed reduction in tumor growth without compromising the efficacy of chemotherapy drugs in a triple-negative breast cancer mouse model, are promising. There are several areas that require further clarification and consideration:

1. While the study demonstrates the beneficial effects of boosting the NAD⁺ metabolome on ovarian function and cancer treatment outcomes, further elucidation of the underlying molecular mechanisms is warranted. Providing insights into how NAD⁺ precursors exert their protective effects on ovarian function and tumor growth would enhance the significance of the findings.

We certainly agree that a detailed mechanistic understanding of how this protection works would greatly benefit this manuscript and the field in general. We spent considerable effort trying to understand this, especially around measures of DNA damage, and the use of small molecule inhibitors – these attempts also relate to point 3, below, and to comments by Reviewer 3.

Unfortunately, we simply could not draw reliable conclusions from those experiments: our immunohistochemistry and western blots for DNA damage response markers, apoptosis and DNA damage itself yielded highly variable results that we consider to be simply unreliable. These experiments failed to show a clear difference from chemotherapy alone, which should act as a positive control for DNA damage and apoptosis. It may be that we did not collect samples at a timepoint that would show these differences, or some other experimental factor, but given the lack of clear answer from these experiments we feel it would not be appropriate to include them. These data are shown on the next page, in answer to a similar question from the reviewer.

To address this point around mechanism, we have added numerous changes throughout the manuscript, most notably a new paragraph in the discussion (pages 16-17).

2. Previous studies have demonstrated a significant impact of chemotherapy drug treatment on ovarian reserve, resulting in extensive apoptosis of primordial follicles. Therefore, the dosage of

chemotherapy drugs plays a crucial role. Hence, the authors need to provide justification for why this particular dosage was chosen.

This is an important point – one can imagine that the degree of protection offered by interventions that raise NAD⁺ will vary according to the dose of the insult. Here, we only tested a single dose for both agents, which were selected simply based on previous work by others – this was mentioned in the methods section...

Doses for doxorubicin and cisplatin (CDDP) were based on previous models for chemotherapy induced infertility (Ben-Aharon et al., 2010, Kerr et al., 2012)

Page 20

To clarify this point, we have added the following sections to the manuscript...

This dose was chosen based on previous work, which established a reduction in follicle counts and demonstrated DNA damage in adult mice (Nguyen et al., 2019).

Page 5

Dox was used at 10 mg/kg, which was chosen based on previous work demonstrating lasting changes to the ovarian reserve (Ben-Aharon et al., 2010).

Page 6

Future work should also aim to test a wider range of chemotherapy doses: it is likely that this intervention will shift the dose response curve of ovarian toxicity, and while our work showed a promising reduction in infertility against previously described gonadotoxic doses (Ben-Aharon et al., 2010; Gonfloni et al., 2009; Kerr et al., 2012; Nguyen et al., 2019), it is unclear whether this protection will translate to higher doses.

Page 17

3. Although the authors attempted to identify potential mechanisms through proteomic analysis, the link between differentially expressed proteins and other factors is not fully established. For example, we know that chemotherapy can induce DNA damage and apoptosis. However, whether NMN supplementation truly affects this process requires further experimental validation.

As above, we absolutely agree that it would have been desirable to find a precise molecular mechanism to explain our results, however the data that we generated to explore exactly this question was inconsistent and inconclusive.

On the point around inducing apoptosis: below are western blots from an experiment where ovaries were maintained *ex vivo* for treatment with doxorubicin, NMN and the PARP small molecule inhibitor XAV-939 – this model was used to ensure ovaries were exposed to each of these agents, and that findings would not be limited by differences in tissue bioavailability. These samples were subject to western blots for cleaved caspase 3 and PARP1, both of which undergo cleavage during apoptosis – we have displayed western blot data here as these data are not subject to selective interpretation based on which IHC section is shown.

As seen here, there is no clear impact of these agents on the cleavage of these proteins. Given that Dox alone should in theory act as a positive control for this assay, we are not confident in including these data in our manuscript or using it to draw any conclusions. The inconclusive nature of this experiment may be related to a technical aspect of the experimental model used, or simply reflect its inherent variability.

4. To clarify the mechanism of action of NMN, we suggest conducting additional experiments such as single-cell transcriptomic sequencing. This will help identify which cells and pathways are altered by NMN supplementation. Such experiments could provide valuable insights for interpreting the results more effectively.

We agree, scRNA-seq would be a valuable technique that could provide a wealth of data to help elucidate a molecular mechanism for this finding. Unfortunately, we are unable to complete this based on its considerable cost – further, our animal ethics approval for chemotherapy treatment has expired and we are unable to readily generate the samples needed for this. We also note and thank the editor for advising that this experiment would not be needed for resubmission. To highlight this excellent suggestion, we have added the following to the end of our new paragraph on limitations and future work in the discussion section:

To better elucidate a potential mechanism that would explain our findings, single cell RNA sequencing should be considered for future work.

Page 17

Referee #2 (Comments on Novelty/Model System for Author):

This is a very nice study, with several lines of investigation carried out at each step, leading to a high level of robustness to the conclusions drawn. The work has the potential to have high medical impact down the line, although at this stage, work is all on the mouse model, or on cell lines.

Referee #2 (Remarks for Author):

This manuscript details the effect of a potential chemo-protectant on female fertility, that of the NADPH system. Overall, it is a very nice, highly robust study. Of particular note, the authors investigate treatment in several ways, and look at two drugs, with effects on ovarian function and fertility also examined in a range of ways, all adding to the strength of the work. It is also clearly written and comprehensively discussed.

We thank the reviewer for their enthusiasm, and appreciate their time in reviewing this paper and providing comments that have helped us to improve this manuscript.

None of my comments are major, but rather aimed at increasing the clarity for readers.

1. Authors should provide a clearer explanation of the statistical analysis carried out, and in particular show results more fully, not just comparing chemotherapy drug-exposed to the chemo + treated

groups. Experiments were designed for 2x2 factorial analysis. Given that, the authors could also show if there was an effect of overall chemotherapy exposure, and of overall NADPH co-treatment, or else they could carry out more post hoc comparisons. As JUST ONE EXAMPLE of this, on p5, the authors state that there is a 'profound' loss of COC induced by DOX. Is that loss significant? What is the no p value?

Apologies we had not shown these analyses in our initial submission, which was to avoid cluttering the appearance of graphs. We have now annotated all figures across this paper to show the results of factorial analyses for chemo, NMN (or NMNAT1/3 overexpression) and their interactions. As specifically requested, we have also added annotations for Bonferroni-adjusted comparisons between the untreated group and the chemo alone groups. Please note that to avoid cluttering graphs, we only annotate these comparisons where the p-value is less than 0.2.

Please note that we have re-ordered the panels in Fig. 1 based on feedback below (also raised by Reviewer 2).

2. More work was carried out examining the effect of doxorubicin than of cisplatin. (Fig 1: NMNAT1 only tested against DOX; and Figs 4-6 work). To be clear, I am not suggesting that yet further work is needed, but just that the authors discuss this, explaining why that was so.

We agree that it would have been ideal to have a set of results showing an equal level of detail not only for both chemotherapy agents, but also for both transgenic mouse strains (NMNAT1 and NMNAT3-Tg) for all measures. We unfortunately were limited by resources, particularly the availability of transgenic animals – in the latter case, we were fortunate enough to use surplus females from another parallel project, thereby reducing animal waste, however were tied to the progress of that project for animals.

We have now added a section to our Discussion that explains this, as per below:

“There were several limitations of our study. As a model for chemotherapy induced infertility we used two chemotherapy agents (doxorubicin and cisplatin), testing three interventions (NMN treatment, NMNAT1-Tg and NMNAT3-Tg overexpression) and using three outcomes for fertility (COC yield, breeding, ovarian histology). It would have been ideal to test every permutation of these insults, interventions and outcomes, however we were unfortunately limited by animal availability and resources. These experiments could be conducted in future work, along with testing a wider range of chemotherapy interventions, including clinically relevant chemotherapy combinations.”

Page 16

3. Figures could be clearer, particularly figures 1-3.

a. Fig 1. The layout could be much clearer, for instance by grouping like panels together. Also, it would be helpful if panels were given headings - e.g. at present is difficult to differentiate between two the different transgenic mouse lines, one small numeral on the x axis.

We agree, and have now rearranged the panels in this figure so that data are arranged in columns for the chemo treatment (left column is doxorubicin, right column is cisplatin), while the rescue interventions (NMN treatment, NMNAT3, NMNAT1) are shown in rows. This is hopefully a more logical order.

We have also added titles to each panel in the plot which should hopefully make this figure easier to read.

b. Figs 2 and 3: it might be worth considering putting the data in panels B and D into a supplemental figure, to allow the key data to be focussed on more.

As suggested, we have moved these panels to Supplementary Figure 2. We are in two minds about this change and happy to be guided by the reviewers and editors as to whether these panels should stay in Supp. Fig. 2, or go back to the main figures, as the manuscript refers to some of the more granular details of the results in these panels, and the goal of generating those panels showing data for individual animals was to improve the transparency of our data presentation. We welcome opinions from the reviewers and the A/Editor on this matter.

c. Fig 4: again the grouping of panels could be clearer. D shows total follicle numbers, but presumably includes data from E and F.

We agree that it would be preferable to show data from antral follicles (panels E and F) directly above the total, but the reason for not doing so was that the current format of panels A-D shows data separated into two panels for “healthy” vs “unhealthy”. Panels E and F are for small and large antrals, respectively, rather than for “healthy” vs “unhealthy” – hence the organisation of panels would become somewhat hard to follow. The values in these panels are in the range of 5-10 follicles, and their contribution to total follicle counts (panel D) is relatively minor, with values in the range of 500-2,000. We believe it is more meaningful to show antral follicle count on its own, as these numbers are more biologically relevant to reproductive function than to total ovarian reserve.

4. The format of references in the text is often odd re listing of authors

This likely refers to the number of authors listed in each reference – we have used the EMBO plugin for referencing style in EndNote, which seems to insert up to the first 10 authors for each paper. Where some papers show the full author list and others show a list of authors followed by “et al”, this likely reflects the number of authors on those papers.

Referee #3 (Remarks for Author):

In the paper "Fertility protection during chemotherapy treatment by boosting the NAD(P)+ metabolome", Wing-Hong Jonathan Ho and co-authors investigated the contribution of dysregulated NAD+ metabolism in chemotherapy-induced female infertility. The authors demonstrated that restoring NAD+ metabolome after chemotherapy treatments improved female fertility without impacting anti-cancer efficacy. The topic presented in the manuscript is interesting and expand our understanding of NAD+ metabolism in chemotherapy-induced infertility with potential translational value. Overall, the data support the conclusions stated by the authors and manuscript is generally clearly written. However, I suggest the following points to be addressed prior to publication:

We thank the reviewer for their positive comments and helpful suggestions, which will significantly benefit this manuscript – please see our responses below.

MAJOR:

- While the authors show that pharmacological (NMN) or genetic (NMNAT1 or 3 tg) interventions to replenish NAD+ are efficacious in improving female mouse fertility, they did not investigate the role of NAD+ consuming enzymes in driving this phenotype. The authors mentioned a couple of times the potential role of PARPs but they did not mention other enzymes such as CD38 which has been reported as overexpressed and key in other Doxorubicin in vivo models (<https://academic.oup.com/circvasres/article/120/3/286/7589532>). Moreover, CD38 has

recently gained attention in the context of ovarian aging (<https://www.sciencedirect.com/science/article/pii/S2589004223020266>, <https://www.nature.com/articles/s43587-023-00532-9>). If other NAD⁺ consuming enzymes are involved in this phenotype, specific pharmacological inhibition of these targets should be considered. The authors should comment and expand their discussion including relevant literature.

This is an excellent point, and we thank the reviewer for highlighting this. We have now altered both the introduction and discussion sections to highlight these recent findings, as follows:

More recently, others have found elevated expression of the NAD⁺ glycohydrolase enzyme CD38 in the ovary during reproductive ageing, and showed that blocking the degradation of NAD⁺ through small molecule inhibition of CD38 can prolong female reproductive lifespan (Perrone et al, 2023; Yang et al, 2024).

Page 4

Chemotherapy can also induce infiltration into the ovaries of immune cells (Du et al, 2022), which can have higher expression of the NAD⁺ consuming enzyme CD38, another determinant of ovarian ageing (Perrone et al., 2023; Yang et al., 2024).

Page 4

One other possibility for altered NAD⁺ flux during chemotherapy treatment could be altered expression of the NAD⁺ consuming enzyme CD38 (Perrone et al., 2023; Yang et al., 2024), the increased expression of which is a cause of doxorubicin induced cardiotoxicity (Peclat et al., 2024), however our proteomics data (Fig. 6) did not show any change in the levels of this enzyme with Dox treatment.

Page 15

Around the role of PARPs in driving this phenotype: we conducted an experiment where mouse ovaries were maintained in *ex vivo* cultures and treated with Dox, NMN and a PARP inhibitor, followed by IHC staining for DNA damage markers and apoptosis. Unfortunately, the results of this were inconclusive – the induction of DNA damage and apoptosis markers was inconsistent, with no real change caused by chemotherapy, let alone a PARP inhibitor and/or NMN. We also performed western blots on ovaries from this experiment, and similarly did not observe a difference in the induction of caspase 3 cleavage – please see WB data above in our reply to Reviewer 1. We are unsure if this relates to our culture conditions or some other aspect of the experimental model, or to the actual lack of impact of *in vitro* doxorubicin treatment on ovaries, but given that these results were so inconclusive, we believe it is safer to exclude them from the manuscript. We certainly agree that further experiments are needed to test for a role of PARP enzymes in this phenotype, and have added the following section to our new paragraph in the discussion on future work:

*Another potential mechanism could be the role of NAD⁺ consuming PARP enzymes which are involved in repairing DNA damage, which could be induced by chemotherapy. Our *in vitro* experiments with PARP inhibitors were inconclusive, and future work should aim to investigate this in more detail.*

Page 15

*While there was a trend towards declining NAD⁺ levels with chemotherapy (Fig. 5C), future experiments should aim to clearly test this possibility, ideally through *in vitro* treatment with small molecule PARP inhibitors.*

Page 17

- Figure 5: the data points show a lot of variability which is expected in these types of analysis. The authors should consider increasing the n to at least n=10.

We agree that it would have been ideal to increase the n values not just in Fig 5 but throughout. Unfortunately, our animal ethics approval for *in vivo* chemotherapy treatment has expired, and re-

obtaining approval in order to generate more samples could take many months. We have therefore added a section to our manuscript to describe this, as follows:

Metabolomics data can have a wide data distribution, and the experiments in Fig. 5 are likely under-powered: once the timeframe of efficacy for NMN treatment can be established, these experiments could be re-examined using higher experimental power.

Page 16

- Figure 5: the authors should comment NMN-induced metabolomic changes such as increased N-Methyl-NAM and decreased NADH.

We had highlighted the change in 1-methyl-nicotinamide levels following NMN treatment in the discussion section as follows:

When the NAD⁺ metabolome is replete and undisturbed, as in young healthy animals that have not received chemotherapy, exogenous NAD⁺ precursors are likely to be excreted as excess surplus, rather than accumulated. NAD⁺ metabolites are excreted through the urine following their conversion to 1-methyl-nicotinamide, and the concept that excess material would be rapidly excreted matches our own data, where NMN treatment drastically increased the production of 1-methyl-nicotinamide (Fig. 5E).

Page 13

To address the reviewer's point, we have also added the following section on the impact of NMN on 1-methyl-Nam in the results section as follows:

As expected, exogenous NMN treatment increased levels of the excretion product 1-methyl-nicotinamide (Fig. 5E), likely reflecting the excretion of excess NAD⁺ precursors.

Page 9

Although there looks to be a trend, there was no impact of NMN on NADH levels (NMN effect $p=0.218$). There was a wide spread of data here, and although we have applied the ROUT method for outlier detection in this manuscript, at the stringency level we have set, the single high reading of NADH in the NMN alone group is not identified as an outlier. To address the reviewer's point, we have also added the following section on the impact of NMN on NADH in the results section as follows:

One rationale for this investigation was that cytotoxic chemotherapy drugs would deplete the NAD⁺ metabolome due to their activation of NAD⁺ consuming PARP enzymes. While there was a trend towards declining NAD⁺ levels with Dox treatment (Fig. 5C), this effect did not reach significance ($p=0.069$), and nor was there an impact of either NMN or Dox treatment on the reduced equivalent NADH (Fig. 5G).

Page 10

- While Figure 5 and 6 are extremely interesting, the metabolomic and proteomic changes observed might not be relevant for the studies showed in Figure 1-4 where mice had longer NMN administration after chemotherapy treatment.

We certainly agree here – although these results could hint at a potential mechanism, we still do not yet know exactly what window of treatment is actually providing the protection against chemo, as our experiments utilised ongoing NMN treatment (or NMNAT1/3 overexpression) during and well after chemotherapy. Given this, we have added the following sections to the discussion...

Another aspect of these experiments that requires further work is in the timing of treatment: we exposed animals to NMN before, during, and long after the chemotherapy insult. While the metabolomics (Fig. 5) and proteomics (Fig. 6) data may provide interesting hints as to potential

mechanism, these samples were taken at an early timepoint only 5 hr after Dox treatment. This period of treatment might not be relevant, and it may be the case that the protection offered by NMN is a long-term, chronic effect that takes place over weeks of treatment. If so, those molecular changes (Figs 5, 6) may be irrelevant to the mechanism of action for NMN treatment. Future work should aim to narrow the window of when NMN treatment offers its protection, to help identify when these molecular changes might be relevant to this phenotype.

Page 16

- Page 10, Supp. Fig.4: This evidence was published last year in the following paper: <https://www.ncbi.nlm.nih.gov/pmc/articles/PMC9818647/> The authors should mention and comment the published data in the context of their study.

Thank you for identifying these references – both the Margier et al paper and the above paper by Peclat et al on CD38 inhibition in doxorubicin induced cardiotoxicity have been added as references in the results section on page 10 and the discussion section on page 15, as follows:

We observed no exacerbation of Dox-induced cardiotoxicity, with a trend towards reduced changes in cardiac function during NMN treatment (Supp. Fig. 4), which is consistent with recent findings using other strategies to elevate NAD⁺ or prevent its breakdown by CD38 (Margier et al, 2022; Peclat et al, 2024; Zheng et al, 2019). Together, these data and other findings suggest that if NMN treatment leads to increased carbonyl reductase activity, this does not translate into an increased risk of cardiac dysfunction.

Page 10

Further, we found that NMN co-treatment with Dox did not increase cardiac dysfunction, which is in line with other recent work testing the impact of administering NAD⁺ precursors, or inhibiting the NAD⁺ consuming enzyme CD38 (Margier et al, 2022; Peclat et al, 2024; Zheng et al, 2019).

Page 15

MINOR:

- Figure 1: it is somehow difficult to follow as the sub-figures are not presented in order (e.g. Fig. 1E mentioned before Fig 1C).

Reviewer 2 also highlighted this point, and we have now amended the order of panels of this figure so that relevant panels are described in sequential order, and are presented visually in a more logical manner, with different chemotherapy separated into columns, and each intervention (NMN treatment, NMNAT1 or NMNAT3 overexpression) shown in rows.

- Figure 7 D and H are low quality and hard to read

Our apologies – this appears to be an issue caused by semi-transparency in the EPS file format when exported from R, we have now re-exported using a different tool which should hopefully have preserved the vector format and make this panels more “crisp” to read. This issue appears to be somewhat dependent on whether the file is being viewed from a Mac or PC, and if the semitransparent lines still appear low quality when viewed from your computer, we hope that the production process will help to address this. For the benefit of the reviewer, we have in the meantime pasted versions of this figure in both EPS and PNG file formats, and hopefully one of them will be clearer.

- I would use the same name in stats legends and axis (e.g. Fig 5: use Dox instead of chemo).

We agree with this comment have changed this label on Fig 5 and throughout.

Figure 7 in **EPS** format (related to comment from Reviewer 3 – see following page)

Figure 7 in PNG format (related to comment from Reviewer 3)

12th Jul 2024

Dear Dr. Wu,

Thank you for submitting your revised study. We have now received the feedback from the 2 referees who evaluated your revised manuscript. As you will see below, while referee #3 is satisfied with the revisions, referee #1 has some concerns on the experimental design (use of whole ovary). We discussed this issue within the team, and agreed that no additional experimental revisions would be required at this time. Therefore, unless you have data at hand that would address this concern, we would like you to adequately discuss this limitation in your manuscript, as mentioned also by the referee. Additionally, please address the following editorial issues:

1/ Manuscript text:

- Please indicate in track changes mode any new modification.
- We can accommodate up to 5 keywords, please adjust accordingly.
- Please move the "For more information" section just before the references.
- The "Methods" section should follow Introduction, results and discussion.
- Methods:
 - o All Materials and Methods need to be described in the main text using our 'Structured Methods' format, which is required for all research articles. According to this format, the Methods section includes a Reagents and Tools Table (listing key reagents, experimental models, software and relevant equipment and including their sources and relevant identifiers) followed by a Methods and Protocols section describing the methods using a step-by-step protocol format. The aim is to facilitate adoption of the methodologies across labs. More information on how to adhere to this format as well as a downloadable template (.docx) for the Reagents and Tools Table can be found in our author guidelines: <https://www.embopress.org/page/journal/17574684/authorguide#structuredmethods>
 - o Animals: please make sure gender and age of the mice are provided for all experiments.
 - o The subsection "Figures" should be a dedicated section, renamed "Graphics"
 - o Thank you for providing the ARRIVE checklist, however please note that it won't be needed for publication. You could alternatively include it in the Appendix file.
- Funding and Acknowledgements: please rename the heading "Acknowledgements". Please make sure that the information provided matches the information entered in the submission system (currently, APP1117538 is missing in the submission system).
- Disclosure statement and competing interests: please remove the sentence on correspondence.

2/ Figures and Appendix:

- Please remove the figures from the manuscript text.
- Please make sure that all figures and figure panels are referenced in the text and in chronological order. Currently, Fig 3A is called out before Fig 2B-G; a callout is missing for Fig 5A,D,F ; there is a callouts for a suppl. fig 6.
- Expanded View Figures should be renamed "Figure EV1", etc. Please add a heading "Expanded View Figure Legends" after the main Figure Legends in the manuscript text.
- Appendix: the two appendix figures should be compiled in a single PDF file and need their legends removed from the manuscript and added to that file, which should be labelled Appendix and uploaded with a table of contents including page numbers.
- Please address the queries of our data editors:

Figure legends:

 1. Please note that the legend for figure 6e-p is missing in the manuscript. This needs to be rectified.
 2. Please note that the exact p values are not provided in the legends of figures 1c-f; 2g; 3e-g; 6e.
 3. Please indicate the statistical test used for data analysis in the legends of figures 5a-h; 6b-p; 7b, h.
 4. Please note that the box plots need to be defined in terms of minima, maxima, centre, bounds of box and whiskers, and percentile in the legends of figures 1b-f; 2e-g; 3e-g; 4b-h; 5a-h; EV 4b.
 5. Please note that information related to n is missing in the legends of figures 7h; EV 4b.
 6. Please note that the measure of center for the error bars needs to be defined in the legends of figures 7h; EV 5a-b.
 7. Please note that scale bar and its definition are missing for figures EV 3a-h.

- 3/ Please provide "The paper explained": EMBO Molecular Medicine articles are accompanied by a summary of the articles to emphasize the major findings in the paper and their medical implications for the non-specialist reader. Please provide a draft summary of your article highlighting
- the medical issue you are addressing,
 - the results obtained and
 - their clinical impact.

This may be edited to ensure that readers understand the significance and context of the research. Please refer to any of our

published articles for an example.

4/ Synopsis:

I slightly edited your text to fit our format, please let me know if you agree with the following or amend as you see fit:

"Female fertility is impaired by chemotherapy in cancer patients. Fertility preservation relies on timely freezing of reproductive material prior to treatment, which is not always feasible. As chemo-induced infertility resembles ovarian ageing, we tested an agent that overcomes reproductive ageing.

- Boosting the nicotinamide adenine dinucleotide (NAD+) metabolome protected against chemotherapy-induced ovarian toxicity and female infertility.
- Effects on fertility were long-lasting, indicating long-term protection of the ovarian reserve.
- The efficacy of chemotherapy against tumour growth in mice was not impaired, indicating that ovarian function protection might not come at the cost of impaired cancer treatment."

Please also suggest a visual abstract to illustrate your article as a TIFF/JPEG/PNG file 550 px wide x 300-600 px high.

5/ As part of the EMBO Publications transparent editorial process initiative (see our Editorial at <http://embomolmed.embopress.org/content/2/9/329>), EMBO Molecular Medicine will publish online a Review Process File (RPF) to accompany accepted manuscripts.

This file will be published in conjunction with your paper and will include the anonymous referee reports, your point-by-point response and all pertinent correspondence relating to the manuscript. Let us know whether you agree with the publication of the RPF and as here, if you want to remove or not any figures from it prior to publication.

I look forward to receiving your revised manuscript.

Yours sincerely,

Lise Roth

***** Reviewer's comments *****

Referee #1 (Remarks for Author):

This reviewer appreciate the detailed response and the efforts made to understand the mechanistic aspects of the study, I have some concerns regarding the experimental design. Specifically, the use of whole ovary samples for proteomics and Western blot validation might not yield conclusive results. This is because chemotherapy drugs may selectively affect a specific subset of ovarian cells that are particularly sensitive to DNA damage. If these sensitive cells constitute a small proportion of the overall ovarian tissue, their response might not be detectable when analyzing the entire ovary. Therefore, the authors could attempt to assess ovarian cell apoptosis in situ on ovary sections. At the very least, this limitation should be mentioned in the discussion.

Referee #3 (Remarks for Author):

The authors addressed the major points raised by the reviewers and improved the manuscript quality. In my opinion, the manuscript is suitable for publication in the current version.

Point-by-point response letter, Ho and Marinova et al (EMM-2024-19836-V3)

We again wish to sincerely thank the Editor and Reviewers for their prompt and positive feedback on this manuscript. We are pleased that we have been able to address most of their previous concerns, and hope that our latest changes to the manuscript will help to highlight the limitation of this study that was mentioned by Referee 1.

In addition to these referee comments, we have addressed each of the editorial comments in our manuscript, and its associated files. This includes a restructuring of the materials and methods section, incorporating a key resources table and structured protocol format, and other changes. We have also uploaded a separate manuscript file with a one-page "The paper explained" summary of this manuscript, also provided at the end of this letter. We hope that these changes address all outstanding comments and requests, and we look forward to your feedback.

Referee #1 (Remarks for Author):

This reviewer appreciate the detailed response and the efforts made to understand the mechanistic aspects of the study, I have some concerns regarding the experimental design. Specifically, the use of whole ovary samples for proteomics and Western blot validation might not yield conclusive results. This is because chemotherapy drugs may selectively affect a specific subset of ovarian cells that are particularly sensitive to DNA damage. If these sensitive cells constitute a small proportion of the overall ovarian tissue, their response might not be detectable when analyzing the entire ovary. Therefore, the authors could attempt to assess ovarian cell apoptosis in situ on ovary sections. At the very least, this limitation should be mentioned in the discussion.

We agree that the lack of cell type-specific data is a problem with the experimental design, both in the unpublished mechanistic data that we provided in our previous response and in the proteomics and metabolomics data shown in the paper. Although we did not include the images, we had also attempted immunohistochemistry for DNA damage and apoptosis markers in a parallel set of ovaries in that experiment, as well as ovaries taken from animals following *in vivo* chemotherapy / NMN treatment. This should have allowed us to gain a better idea around which cell types may have been responsible for the observed phenotype. Unfortunately, we could not reliably observe consistent changes in these markers in any direction, with considerable variability within treatment groups. As with the western blot data in our previous response, we feel that including these data could muddy the field. We have now added a section to the discussion around this point as follows:

To complement this, in situ immunohistochemistry staining for DNA damage and apoptosis markers in ovaries could, at a minimum, provide hints around which cell types are likely to contribute to the observed phenotype. We attempted these studies, but observed considerable variability in staining patterns and intensity within treatment groups, and no treatment effects (data not shown) – as described above, this may relate to an inappropriate selection of timepoints, or other important parameters. Our inability to precisely identify both the molecular mechanism and the relevant cell type(s) that mediate this phenotype are a weakness of this study, and future work should aim to address this in greater detail using in situ immunohistochemistry and single cell sequencing.

Page 17

Referee #3 (Remarks for Author):

The authors addressed the major points raised by the reviewers and improved the manuscript quality. In my opinion, the manuscript is suitable for publication in the current version.

We thank the referee for their efforts and are glad that we have been able to improve the manuscript to their satisfaction.

30th Jul 2024

Dear Dr. Wu,

Thank you for providing your revised files. We are pleased to inform you that your manuscript is accepted for publication and is now being sent to our publisher to be included in the next available issue of EMBO Molecular Medicine!

Please note that I have removed "data not shown" on page 17, as per our journal policy, "data not shown" is not allowed and all data should be included in the manuscript. As you detailed the experiments and inconclusive results in your rebuttal letter, and to some extent in the manuscript, there is no requirement for showing the data here.

Yours sincerely,

Lise Roth
